# Large Language Models as Commonsense Knowledge for Large-Scale Task Planning

**Zirui Zhao**    **Wee Sun Lee**    **David Hsu**
National University of Singapore
`{ziruiz, leews, dyhsu}@comp.nus.edu.sg`

## Abstract

Large-scale task planning is a major challenge. Recent work exploits large language models (LLMs) directly as a *policy* and shows surprisingly interesting results. This paper shows that LLMs provide a *commonsense model* of the world in addition to a policy that acts on it. The world model and the policy can be combined in a search algorithm, such as Monte Carlo Tree Search (MCTS), to scale up task planning. In our new LLM-MCTS algorithm, the LLM-induced world model provides a commonsense prior belief for MCTS to achieve effective reasoning; the LLM-induced policy acts as a heuristic to guide the search, vastly improving search efficiency. Experiments show that LLM-MCTS outperforms both MCTS alone and policies induced by LLMs (GPT2 and GPT3.5) by a wide margin for complex, novel tasks. Further experiments and analyses on multiple tasks—multiplication, travel planning, object rearrangement—suggest *minimum description length* (MDL) as a general guiding principle: if the description length of the world model is substantially smaller than that of the policy, using LLM as a world model for model-based planning is likely better than using LLM solely as a policy.[1]

## 1 Introduction

Consider, for example, an autonomous robot butler in a household environment. The human user sits in the living room and asks the robot to "Put fruits into the fridge." The robot looks for fruits, such as apples, peaches, etc., which may be on a table in the dining room, on the kitchen counter, but unlikely in a wardrobe in the bedroom. To complete the task, the robot has to consider fruits' likely locations as well as their spatial relations, traverse these locations efficiently, and finally put them into the fridge. A household environment typically contains hundreds of movable items and locations, resulting in a huge search space that makes the task very challenging for the robot.

Recently, multiple attempts exploiting pre-trained large language models (LLMs) show surprisingly interesting results for such tasks [22, 18, 2, 19, 5, 44]. Their underlying idea is simple: treat the LLM as a policy and query it directly for the next actions, given the history of past actions and observations. We call this strategy *L-Policy*, which exploits LLMs' vast commonsense knowledge to circumvent the challenge of searching a very large space. In our example task, L-Policy may simply instruct the robot to move to the hallway and then to the kitchen. Even though LLMs are trained on internet-scale data, this strategy shows limits in generalization [10, 4], especially when encountering uncommon, complex tasks. Alternatively, we may use LLMs' knowledge to build a world model and apply a planning algorithm to the model. The world model may contain, e.g., a belief over the target object's location, which biases the search and drastically improves search efficiency. We call this strategy *L-Model*. L-Model's performance depends on two critical preconditions: the accuracy of the world model and the efficiency of the planning algorithm. The former is a question of sample complexity for learning, and the latter is that of computational complexity.

---

[1]The code and supplementary materials are available at `https://llm-mcts.github.io`.

37th Conference on Neural Information Processing Systems (NeurIPS 2023).

This paper presents LLM-MCTS (shown in Fig 1), which combines the ideas of L-Model and L-Policy for large-scale task planning. Like L-Model, LLM-MCTS uses an LLM to build a commonsense world model; it then uses the model to perform Monte Carlo Tree Search (MCTS) [8] online for the next actions. During the tree search, LLM-MCTS chooses the promising action branches heuristically by querying the LLM. This is similar in spirit to L-Policy. While L-Policy commits to the actions chosen by the LLM for execution, LLM-MCTS uses these choices only as a search heuristic.

We evaluate LLM-MCTS in VirtualHome [24], a standard household activity simulation platform widely used in earlier work [18, 22, 25, 33]. The evaluation set consists of 800 randomly generated large-scale, partially observable object rearrangement tasks. In each task, a robot aims to fetch a common household item with an unknown location and place it in a designated container. Our main experimental findings are summarized below:

F1. *L-Model performs poorly.* There are two possible reasons. One is model inaccuracy: the robot has an incorrect belief of the target object's location. The other is huge search space size, beyond the reach of even the state-of-the-art MCTS algorithm. Further experiments indicate that search space size is the main cause.

F2. *L-Policy performs reasonably with both GPT2 and GPT3.5, but the performance degrades quickly for novel, complex tasks.* This generally corroborates with earlier results [22, 18, 2, 19].

F3. *LLM-MCTS outperforms L-Model.* This clearly shows the benefit of using the LLM as a heuristic policy to guide the search.

F4. *LLM-MCTS outperforms L-Policy, especially for novel, complex tasks.* LLM-MCTS basically combines L-Model and L-Policy. Since the L-Model performs very poorly on its own, why does the combination outperform L-Policy? One explanation is that search space size is the main cause of L-Model's poor performance. The LLM-induced world model is sufficiently accurate; tree search with this world model, when limited to the neighbourhood of the LLM-induced policy, provides improved performance over L-Policy.

The explanation for (F4) begs a new question: with an efficient planning algorithm for large search space, *would L-Model outperform L-Policy?* To answer this question, we study two related, simpler tasks: multiplication of two large numbers and multi-hop travel planning. We discuss multiplication here and travel planning in Section 4.1. A decimal number is described as a sequence of $n$ digits, $(d_{n-1}, d_{n-2}, \ldots, d_0)$. There are two methods of implementing multiplication with an LLM. The first one corresponds to L-Policy. We represent the multiplication function as a table. Each row or column corresponds to a number. The table entry is the multiplication of two numbers, obtained by querying an LLM. Experimentally, GPT4 performs single-digit multiplication perfectly with $100\%$ accuracy, 2-digit multiplication with $99\%$ accuracy, 4-digit multiplication with merely $4\%$ accuracy, and fails almost completely on 5-digit multiplication [10]. The second approach uses LLM-derived small single-digit multiplication tables, which GPT4 performs with $100\%$ accuracy. To multiply multi-digit numbers, it applies the long multiplication algorithm with the single-digit table. This method corresponds to L-Model. While long multiplication differs from planning in algorithmic details, it plays the same role in L-Model and is highly efficient. Clearly, this second method achieves $100\%$ accuracy for arbitrarily large numbers, provided that the single-digit multiplication table is accurate. So, the L-Model outperforms L-Policy for the multiplication task, contrary to the finding for object rearrangement tasks.

*How do we choose between L-Model and L-Policy then?* One idea is the minimum description length (MDL) principle. Theoretical analysis suggests that a hypothesis with a shorter description length has a smaller generalization error and is preferred [28]. For multiplication, the method corresponding to L-Policy uses a large table of $O(10^{2n})$ entries. It takes $O(n10^{2n})$ bits to represent it. The method corresponding to the L-Model uses a single-digit multiplication table of constant size. The long multiplication algorithm can be encoded in any reasonable programming language with constant size. So, the total representation size is constant. According to MDL, the L-Model has a smaller generalization error than the L-Policy for multiplication, with sufficiently large $n$. This is fully consistent with experimental results [10]. The analysis of travel planning provides further evidence (Section 4.1).

In summary, LLM-MCTS combines the ideas of L-Model and L-Policy, outperforming either alone for complex task planning, particularly, object rearrangement (Sections 2 and 3). To choose between L-Model and L-Policy, MDL provides a useful guiding principle (Section 4). In essence, simplicity is preferred, a well-known general principle in machine learning.

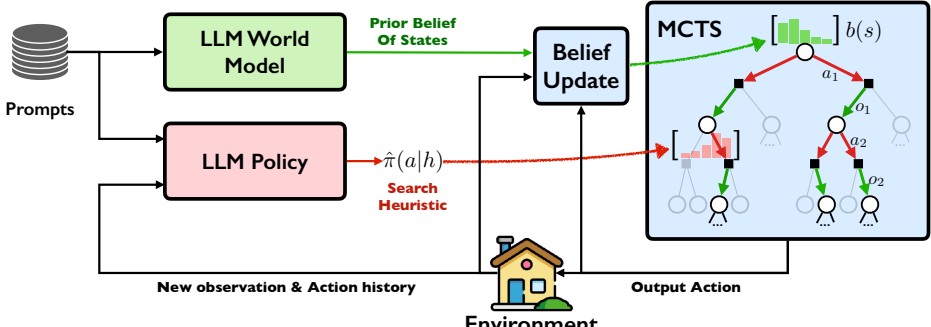

Figure 1: Overview of LLM-MCTS. For each simulation in the MCTS, we sample from the commonsense belief to obtain an initial state of the world and use the LLM as heuristics to guide the trajectory to promising parts of the search tree.

## 2 LLM-MCTS: Monte Carlo planning with commonsense knowledge

We aim to solve task-planning problems in large-scale domains with partial observation. One example is object rearrangement tasks [3] in household environments. It is a meaningful and challenging problem with a large-scale and long-term planning horizon. It has many practical implications in everyday life [3, 34, 39, 20, 17], such as setting the table, tidying up the room, loading the dishwasher, etc.. To solve the problem, we present LLM-MCTS (shown in Fig 1), which combines the L-Model and L-Policy for large-scale planning. It uses LLM to build a commonsense world model to perform MCTS for reasoned planning and uses L-Policy to guide the MCTS and reduce the large search space.

### 2.1 Task planning

We focus on task-planning problems with partial observation and large-scale domains. The problem can be formulated as a Partially Observable Markov Decision Process (POMDP): $(S, A, \Omega, T, O, R, \gamma)$. The state space $S$ define the state of the robot and its environment. The action space $A$ defines the action that the robot can do. $\Omega$ is the observation space. $T$ defines the transition function of states, which we assume to be given. $O$ is the observation function that provides partial information about a state. $R(s, a)$ is the reward function determined by the action $a$ taken at the state $s$. The discount factor is specified by $\gamma$. The history trajectory $h_t$ at time step $t$ consists of a sequence of executed actions and received observations up to time $t - 1$, $h_t = (o_0, a_0, o_1, a_1, \ldots, o_{t-1}, a_{t-1})$. The objective is to find an optimal policy $\pi^*(h_t)$ that maximize the expected cumulative rewards $\pi^*(h_t) = \arg\max_{a \in A} \mathbb{E}\left[\sum_{i=0}^{\infty} \gamma^i R(s_{t+i}, a_{t+i}) | a_t = a\right]$.

In this work, we focus on the object rearrangement task, though our approach is a general method for large-scale task planning. Object rearrangement is a representative embodied AI task [3, 34, 39, 20, 17] with various daily applications, such as setting the table, tidying up the room, loading the dishwasher, and more. It is a challenging task [3] as the robot must navigate, locate target objects and positions, and execute multi-step planning. As with hundreds of items and containers in the domain, identifying target objects can be challenging. Even with known state information, it requires long-horizon planning to achieve the goal. In addition, the vast number of domain objects leads to a large action space, given the actions are to interact with objects. The vast action space produces an exponentially large search tree, making the planning extremely challenging.

Following the approach of [22, 25, 33], we model the task as a POMDP as outlined above. The state $S$ comprises variables denoting the positions of the robot, movable items, and containers. Actions $A$ encompass five predefined actions from VirtualHome, parameterized by object/container/rooms: (1) *pick(object)*, where the robot collects an observed, proximate object; (2) *place(object,placement)*, allowing the robot to set a picked object nearby or inside an open container; (3) *open(container)* and (4) *close(container)*, for interacting with an observed, nearby containers; and (5) *move(room/object/container)*, where the robot relocates within a room or near an observed object/container. Given our assumption of the robot's familiarity with house structures and the manageable size of the house, the robot can move directly to designated rooms. The deterministic transition $T$ is pre-defined by actions. Partial observation $O$ enables the robot to discern object/container positions within its room or an opened container at its location. The objective is to reorganize household items based on verbal instructions, represented by a reward of $R$ for achieving the desired item arrangement.

## 2.2 LLM as a commonsense world model

A commonsense prior belief of states can improve the effectiveness of object and location searches by prioritizing the search to appropriate locations. Our approach utilizes LLM's commonsense knowledge to generate the initial belief of states, which is updated with each action and observation in the real world. MCTS samples from the belief in simulation to estimate the value of the action.

**Initial belief of state.** We use object-centric state representation and categorize the objects in the house as moveable objects (e.g., apples), containers (e.g., fridge), and surfaces (e.g., kitchen table). The states of a moveable object might be inside the containers or on the surfaces. The containers and surfaces should be inside a room. Similar to [22, 25], we maintain the belief in object-centric graphs, where nodes are objects and edges describe abstract-level relationships (e.g., apples are inside fridge, fridge is inside kitchen) between objects and rooms. Details are in the Appendix C.2.

Assume a dataset $\mathcal{D}$ is accessible, containing expert actions and observations in similar household environments to solve daily tasks. LLMs can use the observations in the data to know what are the objects in the house and predict their positions, forming the commonsense belief of the state. To achieve this, we find all the objects, containers, and surfaces that appeared in the dataset $\mathcal{D}$ to form a list of objects $\mathcal{D}_{\mathrm{obj}}$ using a unique name for all of them. To approximate $b(s_0)$, we ask the LLMs to sample the positions of objects $M$ times. For each sample, we ask the LLM to predict the position of objects using $\mathcal{D}_{\mathrm{obj}}$ and a fixed prompt. For instance, we ask LLM to complete "*The containers in the apartment are: fridge, ...; The surfaces in the apartment are: kitchen counter, ...; Question: what are the possible positions of strawberry? Answer: inside fridge, inside pantry... Question: what are the possible positions of apple? Answer:__.*" We use three prompt examples to provide example formats of the response. The exact prompts we used are provided in the appendix. As the responses from LLM are free-form natural language, we have to precisely map those expressions to $\mathcal{D}_{\mathrm{obj}}$ for consistent state representation. Thus, we encode the names of objects in the LLM's response into embeddings using sentence-BERT $f(\cdot)$ [26] and examine their cosine similarity to the unique name of objects in $\mathcal{D}_{\mathrm{obj}}$: $\mathrm{CosineSim}(e_i, e) = \frac{f(e_i)f(e)}{\|f(e_i)\|\|f(e)\|}$, where $e$ is the name of objects, containers, or surfaces in the LLM's response, and $e_i \in \mathcal{D}_{\mathrm{obj}}$ are the unique names in the object list. We select the most similar expressions in $\mathcal{D}_{\mathrm{obj}}$ to form the sampled state. For example, when querying the position of an apple, the LLM's response is "on the kitchen table," we use the above technique to translate "the kitchen table" to "kitchentable," a unique name in $\mathcal{D}_{\mathrm{obj}}$.

**Goal.** Similar to [40], we use LLMs to translate the natural language goal into a formal goal for MCTS. We use a fixed set of prompt examples for LLM to interpret natural language goals, such as "put one apple into the fridge" is translated as a tuple "(*apple*, *inside*, *fridge*)." For compositional instructions, it will translate it into multiple tuples, such as "put one apple on the kitchen table and one plate inside the dishwasher" is translated as "(*apple, on, kitchentable*), (*plate, inside, dishwasher*)." We precisely map the LLM-generated goal into the admissible expressions in $\mathcal{D}_{\mathrm{obj}}$ for search using the same representation as the state. In MCTS, the goal is used to identify the reward. As the representations are the same, we can directly check whether the object's state is the same as the goal by string matching. If the goal is reached, it will receive a large positive reward, or 0 otherwise.

## 2.3 LLM as a heuristic policy

We use LLMs to play the role of $\pi(a|h)$ in PUCT to guide the action selection in the simulation procedure. In this procedure, the LLM takes as input the examples in the dataset, the goal description, the current observation, and the history of actions, and then outputs the suggested action plan (e.g., "*Next actions: move to the kitchen, open the fridge, ...*"). Similar to [22], the observations and goal description are translated into English sentences. As the answer of LLM is from the conditional distribution of the following words given the context, it can also be viewed as a commonsense policy of actions to take conditioned on the context of tasks, observations, and completed actions. However, direct implementation and access to the probability value of the GPT-3.5 is not available. Thus, we propose an empirical policy distribution $\hat{\pi}$ that uses sampling to approximate the policy distribution.

We sample the LLM for $M$ times to approximate the policy probability distribution. For each sample, we query the LLM with prompt and trajectory history $h$ and receive an answer of the following actions to take $\alpha_i \sim \mathrm{LLM}(h, \mathrm{prompt})$, where $\alpha_i$ is the first action of the answer. The prompt examples are retrieved from the dataset according to the similarity to the current language instruction $\ell$. We use [26] to translate the instructions in the dataset $\ell_i \in \mathcal{D}$ into embedding and examine their

**Algorithm 1** LLM-MCTS

```
 1: procedure SEARCH(h, b, T, N)           20:      if γ^d < ε or done = True then
 2:     n ← 0                              21:          return 0
 3:     while n < N do                     22:      end if
 4:         s ~ b(s)                       23:      if h is not in T then
 5:         SIMULATE(s, h, False, 0, T)    24:          T ← T ∪ h, N(h) ← 0
 6:         n ← n + 1                      25:          ∀a ∈ A, N(h, a) ← 0, Q(h, a) ← 0
 7:     end while                          26:          return ROLLOUT(s, h, done, d)
 8:     return argmax_{a∈A} Q(h, a)        27:      end if
 9: end procedure                          28:      π̂(a|h) ← QUERYLLMPOLICY(h)
10: procedure ROLLOUT(s, h, done, d)
11:     if γ^d < ε or done = True then     29:      a* ← argmax_{a∈A} Q(h,a)+cπ̂(a|h) (√N(h))/(N(h,a)+1)
12:         return 0
13:     end if                             30:      (s', o, r, done) ~ G(s, a*)
14:     a ~ π_rollout(h, ·)                31:      h' ← PUSHBACK(h, [a*, o]), d' ← d+1
15:     (s', o, r, done) ~ G(s, a)         32:      R ← r + γ·SIMULATE(s', h', done, d', T)
16:     h' ← PUSHBACK(h, [a*, o]), d' ← d+1 33:     N(h, a*) += 1, N(h) += 1
17:     return r + γ·ROLLOUT(s, h', done, d') 34:   Q(h, a*) ← Q(h, a*) + (R−Q(h,a*))/(N(h,a*))
18: end procedure                          35:      return R
19: procedure SIMULATE(s, h, done, d, T)   36: end procedure
```

cosine similarity to the current instruction: $\text{CosineSim}(\ell_i, \ell)$. In experiments, we use a subset of $\mathcal{D}$ to show its performance when restricted to a small training set. We select the top $K$ similar instructions and use the corresponding expert trajectories as a $K$-shot prompt. However, the answer $\alpha_i$ is a free-formed natural language sentence that cannot be mapped to admissible actions for the agent directly. To ensure that the action can be executed, we follow the method in prior works [18] to represent the actions and admissible actions by embeddings from [26] and evaluate their cosine similarity $\text{CosineSim}(\alpha_i, a)$. The empirical policy distribution is formulated as follows: $\hat{\pi}(a|h) = \lambda \frac{1}{|A|} + (1 - \lambda)\text{Softmax}\{\sum_{i=1}^{M} \text{CosineSim}(\alpha_i, a) - \eta\}$, where $\eta$ is the average value of $\sum_i \text{CosineSim}(\alpha_i, a)$ and $|A|$ is the size of the admissible action space. $\lambda$ is a hyper-parameter that adds randomness to the belief, as the sampled actions from LLM could be very deterministic. Therefore, the empirical policy distribution is a mixture of approximated policy from LLM and uniform distribution. The example prompts are provided in Appendix F.

## 2.4 Monte Carlo tree search

We integrate the commonsense world belief and policy from LLM in MCTS, presented in Alg 1. For each simulation, MCTS samples a state from the belief $b(s)$ at the root (line 4). It independently samples one position for each object to construct a state $s$. This sampled state $s$ is then employed in the simulation, generating a new tree trajectory. An action $a^*$ is chosen during the simulation based on the $Q$ value, visit counts, and LLM policy (lines 28 and 29). The observation and transition function, denoted as $\mathcal{G}$ (lines 15 and 30), predict the next state $s'$ given the selected action $a^*$ and the sampled state $s$, thus progressing to the subsequent step in the simulation (lines 30 and 31). When encountering leaf nodes in the tree, MCTS expands the tree and performs a random rollout for the corresponding node (lines 23 to 26). A uniform policy is employed to sample actions in the rollout, and the discounted reward is then returned (lines 14 to 17). Upon completing the task or reaching the maximum depth, the accumulated rewards are backpropagated, updating each node's estimated $Q$ value (lines 32 to 35). Following $N$ simulations, the output action is determined based on the estimated $Q$ value (lines 3 to 8). Upon completion of the search process, the agent will execute an action and receive a new observation. For simplicity, we assume that the observation and transition functions are deterministic and known. In cases where an object is detected, its corresponding position within the belief will be updated with the observed position. Conversely, if the object remains undetected at certain positions, the belief regarding its presence in those positions will be rendered null, denoted by a zero value.

# 3 Experiments

## 3.1 Experimental setup

**VirtualHome.** We proceed with our experiments in the VirtualHome [24], a large household simulated environment with a large domain, partial observations, and large action space. It contains hundreds of interactive items and containers with various types of rooms. It is a well-suited platform for evaluating embodied decision-making for solving daily tasks in household environments.

**Data.** To generate data for prompting and baseline training, we follow [25] to create 2000 tasks with randomly initialized scenes and expert trajectories. There are several settings for the evaluation. *Simple* tasks are the tasks that only require the rearrangement of one item generated from the same distribution as the training dataset. *Comp.* refers to the composition of simple tasks in order to rearrange multiple objects sampled from the same distribution as the dataset (e.g., "Put plate on kitchen table and chicken inside fridge" is the composition of "put plate on kitchen table" and "put chicken inside fridge," ). The composition of tasks increases the planning horizon, making it more challenging to complete. In evaluation, we also use the *Novel Simple* tasks with seen items (e.g., in the dataset, we have "put one plate on the kitchen table" and "put one chicken inside the fridge," and we use "put one plate inside the fridge" and "put one chicken on the kitchen table" to evaluate; these tasks are not included in the training dataset). For compositional tasks, we include *Novel Comp*ositional tasks, with 2 or 3 primary tasks composed, denoted as *NovelComp(2)* and *NovelComp(3)* (e.g., we have "put plate on kitchen table" and "put chicken inside fridge," in dataset but their composition "Put plate on kitchen table and chicken inside fridge" is not.) We also generate scenes at a *Novel Apartment* for testing, where the distribution of object positions differs from the dataset.

The expert data are generated by an Oracle agent implemented in [25]. The expert has the full knowledge of the environment (hence, does not need to understand where objects are likely to be placed) and uses handcrafted heuristics for completing various tasks. It uses regression planning to search for solutions to a task. We collect the actions and observations of the expert completing the tasks in the VirtualHome simulator as the dataset. There are 10,000 trajectories in total for training the baseline. To show the capability of LLMs when using a small training set, we only select 200 instances uniformly at random from the dataset as prompt candidates for the LLM model and policy when used in a few-shot mode with no fine-tuning. We also generated 800 tasks in total for evaluation.

**Evaluation.** We evaluate the success rate of completing the tasks within 30 steps, while a typical task can be finished within at most 15 steps. The task is considered successful if all the requirements of object positions are satisfied. For example, given the instruction "Put one apple inside the fridge," the task is successful if any apple is in the fridge. For simplicity, we don't consider the task of rearranging a very specific object, e.g., putting the leftmost apple in the fridge.

**Baselines.** We evaluate several baselines to compare. *UCT* [21]: We use the UCT algorithm to conduct planning without commonsense knowledge and use the ground-truth reward function in simulation. We use uniform distribution as the initial belief for states of objects. It is to provide evidence that commonsense knowledge improves planning efficiency. *Finetuned GPT2 policy* [22]: we use the training dataset with 10000 trajectories to fine-tune GPT-2 as the planning policy. This is to show that larger pre-trained LLM without fine-tuning outperforms the smaller model fine-tuned in specific tasks. *GPT3.5 Policy* [18]: LLM takes as input the instructions and history of actions and the currently visible objects to generate the next action. We use the LLM as the policy only, with a few examples as prompts to interact with the environments. This baseline demonstrates the benefits of additional information from the commonsense model and algorithmic benefits from MCTS.

## 3.2 Results

**Main result.** The main results of the experiments are shown in Table 1, reporting the success rate of our method

Table 1: Main results: mean ± standard error of success rate (%)

| Method | Simple | Comp. | NovelSimple | NovelComp.(2) | NovelComp.(3) |
|---|---|---|---|---|---|
| | | | Seen Home | | |
| UCT [21] | 0.0±0.0 | 0.0±0.0 | 0.0±0.0 | 0.0±0.0 | 0.0±0.0 |
| finetuned GPT2 policy [22] | 81.3±2.4 | 59.0±6.7 | 41.2±7.1 | 30.9±2.8 | 2.3±1.5 |
| GPT3.5 Policy [18] | 83.4±6.8 | 47.0±7.8 | 74.3±4.0 | 48.2±8.8 | 5.4±2.0 |
| GPT3.5-MCTS (Ours) | 91.4±3.3 | 71.2±6.2 | 88.1±4.3 | 72.6±6.9 | 33.6±3.1 |
| | | | Unseen Home | | |
| UCT [21] | 0.0±0.0 | 0.0±0.0 | 0.0±0.0 | 0.0±0.0 | 0.0±0.0 |
| finetuned GPT2 policy [22] | 65.5±3.4 | 39.9±5.2 | 33.4±6.4 | 12.8±3.9 | 1.1±0.9 |
| GPT3.5 Policy [18] | 74.3±5.0 | 43.3±4.0 | 67.8±4.9 | 54.0±3.0 | 6.9±2.1 |
| GPT3.5-MCTS (Ours) | 82.9±3.2 | 71.9±5.6 | 79.3±3.3 | 70.4±6.4 | 38.8±3.4 |

and baselines in completing the tasks in VirtualHome environments. In this result, GPT3.5-MCTS outperforms all the compared baselines, especially for unseen situations. UCT works poorly in all conditions, as the poor model and the huge search tree make the planning intractable. Thus, we focus our discussion on comparing the finetuned GPT2 policy and GPT3.5 policy. For *Simple*, in-distribution tasks, the planning horizon is relatively short. Finetuned GPT2 policy, GPT3.5 Policy, and our method work reasonably well, but our method still outperforms the baselines. For *Novel Simple* tasks, finetuned GPT2 policy works significantly worse than GPT3.5 Policy and GPT3.5-MCTS. This is because the fine-tuning of narrow tasks results in a biased distribution of the policy and compromises generalizability. GPT3.5 Policy and GPT3.5-MCTS work better due to the LLM's few-shot planning capability. GPT3.5-MCTS works better for both situations. It benefits from the MCTS' look-ahead search that explore possible states for potential outcomes in order to make reasoned decisions.

For the *Comp*ositional, in-distribution tasks, the finetuned GPT2 policy and GPT3.5 policy get significantly worse performance, while GPT3.5-MCTS works far better. The finetuned GPT2 policy is trained by behavior cloning that suffers from compounding errors. Therefore, when the planning horizon gets longer, the influence of the errors accumulates and compromises the overall performance significantly. As for GPT3.5 Policy, the longer horizon potentially introduces more errors during planning, which might not be included in the prompt examples. Without suitable guidance from prompt, we cannot guarantee the GPT3.5 Policy will carry out suitable replanning when encountering errors. MCTS encourages exploration to a certain extent of different possible actions during searching, introducing additional guidance to the GPT3.5 policy to look into other possible solutions. This is because the action selection procedure in GPT3.5-MCTS is not purely determined by GPT3.5 Policy but also by the $Q$ value and visit counts. Thus, MCTS encourages GPT3.5 Policy to explore other possible search directions instead of excessively applying certain actions sampled by itself.

**Ablation study.** We conduct ablation studies to see the individual contributions of different components within the GPT3.5-MCTS framework. The *No Heuristic Policy* version of GPT3.5-MCTS refers to the absence of PUCT guided by the GPT3.5 Policy for action selection. Instead, it solely relies on UCT with an initial commonsense belief derived from LLM. The variant employing the *Uniform State Prior* utilizes a uniform prior belief regarding states, in contrast to the LLM-generated initial belief employed during the search process. Lastly, the variant operating in a *Fully Observable* environment aims to assess the accuracy of LLM's knowledge in modeling the world.

Table 2 shows the results of our ablation study. The outcomes obtained under the *No Heuristic Policy* version highlight the significance of heuristic policies in facilitating MCTS to conduct efficient searches for complex and large-scale planning tasks. Con-

Table 2: Ablation Study: mean $\pm$ standard error of success rate (%)

| Method | Seen Home | | | | |
| --- | --- | --- | --- | --- | --- |
| | Simple | Comp. | NovelSimple | NovelComp.(2) | NovelComp.(3) |
| GPT3.5-MCTS (No Heuristic Policy) | 0.0±0.0 | 0.0±0.0 | 0.0±0.0 | 0.0±0.0 | 0.0±0.0 |
| GPT3.5-MCTS (Uniform State Prior) | 3.2±1.1 | 0.0±0.0 | 1.1±0.4 | 0.0±0.0 | 0.0±0.0 |
| GPT3.5-MCTS (Fully Observable) | 94.0±2.1 | 80.7±3.3 | 94.3±2.4 | 78.5±4.0 | 34.0±4.4 |
| GPT3.5-MCTS (Ours) | 91.4±3.3 | 71.2±6.2 | 88.1±4.3 | 72.6±6.9 | 33.6±3.1 |
| Method | Unseen Home | | | | |
| | Simple | Comp. | NovelSimple | NovelComp.(2) | NovelComp.(3) |
| GPT3.5-MCTS (No Heuristic Policy) | 0.0±0.0 | 0.0±0.0 | 0.0±0.0 | 0.0±0.0 | 0.0±0.0 |
| GPT3.5-MCTS (Uniform State Prior) | 1.1±0.2 | 0.0±0.0 | 0.0±0.0 | 0.0±0.0 | 0.0±0.0 |
| GPT3.5-MCTS (Fully Observable) | 85.1±5.0 | 77.5±3.2 | 82.2±3.3 | 76.6±3.1 | 37.9±2.9 |
| GPT3.5-MCTS (Ours) | 82.9±3.2 | 71.9±5.6 | 79.3±3.3 | 70.4±6.4 | 38.8±3.4 |

versely, the results of the *Uniform State Prior* row indicate that incorrect world models compromise search performance. This is because the model of the world determines the $Q$ value. The wrong model results in an inaccurate estimation of the $Q$ value, misleading the search process toward irrelevant locations. The *Fully Observable* results demonstrate that GPT3.5-MCTS with perfect knowledge of the environment only slightly outperforms its counterpart without it, implying that the commonsense knowledge of LLM regarding world modelling suffices for practical purposes.

**Failure analysis.** Policy, model, and translation errors are the primary causes of failures. Among these, policy errors are responsible for the majority of the failures. Oftentimes, the policy produces unreasonable behaviours that mislead the search procedure. For example, it usually outputs inadmissible actions, such as "*walk to the cutleryfork*" where the "*cutleryfork*" is not in the observation. It also produces back-and-forth behaviours, resulting in an unreasonable heuristic and slowing the search procedure. For example, when putting objects inside the microwave, it is sometimes struck by repeatedly opening and closing the microwave. For model error, the predicted positions of objects are not always correct. Since a random rollout policy is employed, incorrect object states can result in higher $Q$-values than correct states, leading to misguided exploration. The wrong translation also

compromises the performance as we translate the response from LLM to admissible action or object names to ensure executability. This is partly caused by the VirtualHome environments, as the policy might not understand the underlying logic of the actions in VirtualHome, such as you have to walk close to interact with the object. Thus, if the LLM outputs "open fridge" but is not close enough to the fridge, the action will be translated to other admissible actions ("open fridge" is not inside the admissible actions for this case as it is invalid due to the setting of VirtualHome).

# 4 LLM as a model or a policy?

When would using LLM as a model outperform using LLM as a policy, and vice versa? We propose using the minimum description length (MDL) principle, also known as Occam's Razor from the philosophy of science, to gain insights into the issue. The MDL principle suggests choosing the method that has a shorter description when both methods fit the training data well. MDL has been formalized in various ways. One formal statement (from section 7.3 of [28]) is provided here:

**Theorem 4.1** (Occam's Razor). *Let $\mathcal{H}$ be a hypothesis class and let $d : \mathcal{H} \to \{0, 1\}^*$ be a prefix-free description language for $\mathcal{H}$. Then, for every sample size, $m$, every confidence parameter, $\delta > 0$, and every probability distribution, $D$, with probability greater than $1 - \delta$ over the choice of $S \sim D^m$ we have that, $\forall h \in \mathcal{H}, L_D(h) \leq L_S(h) + \sqrt{(|h| + \ln(2/\delta))/2m}$ where $L_S(h)$ is the empirical loss of $h$ on the $S$, $L_D(h)$ is the expected loss of $h$, and $|h|$ is the length of $d(h)$.*

According to Theorem 4.1, we can bound the expected loss of a solution $h$ by the description length $|h|$ and the training loss $L_S(h)$. We do not know the LLM training loss for using it as a model or as a policy, but for the purpose of gaining insights, it is reasonable to assume that they are both small. In that case, the MDL principle suggests selecting between a model or policy depending on which of them has the smaller description length given the description language.

Numerous caveats should be observed when using Theorem 4.1 to gain insights into the behaviour of LLM as a model and policy. The theorem assumes that the training data is independent and identically distributed (iid), which is likely not true in practice. Nonetheless, the qualitative behaviour is often similar for non-iid data. As the training data of GPT is unknown, when comparing the description length, we also assume that the training data for each subproblem is roughly the same. In addition, using the theorem to gain insights requires major assumptions on the predictor classes $\mathcal{H}$ for model and policy; here, we assume that $\mathcal{H}$ is the model (or policy) class that we are analysing and assume that LLM training using powerful approximators such as transformers has similar behaviour to training using the model (or policy) class. Finally, depending on how the model and policy are used, error propagation may need to be analysed separately.

Besides the multiplication example described earlier, we discuss an air travel planning task and the VirtualHome object rearrangement task examined earlier.

## 4.1 Travel planning

Consider planning for air travel from a starting city to the destination city. To solve the problem through L-Model, we need to have the model – the direct flights out of each city – together with a shortest path algorithm. The model can be represented as a graph, which is likely to be sparse in the real world. For a sparse graph, an adjacency list will give a compact representation. Assuming that the total number of edges grows proportionally to the number of cities, $O(n \log n)$ bits would be sufficient to describe a graph with $n$ cities, with approximately $\log n$ bits used to describe each city in the adjacency list structure. The shortest path algorithm can be described by any reasonable programming language with a constant size. The method regarding L-Policy can be represented as a 2-dimensional table, where each row and column denotes the current and destination city, and the table entry describes the next city to fly to on the shortest path from the current city to the destination. The next city in the table can be described with $\log n$ bits. As such, with $n$ cities in the rows and columns, there should be approximately $n^2 \log n$ bits in total. Thus, according to MDL, the L-Model has a shorter description length and should make fewer generalization errors than the L-Policy for travel planning with sufficiently large $n$.

**Experiment.** We conducted experiments about planning for air travel from a starting city to a destination city, which we analyzed above. We utilized GPT-3.5 to generate flight paths between cities. We compare it to the GPT-3.5 model-based approach: we use GPT-3.5 to predict neighbouring

cities connected by a direct flight, which feeds into the uniform-cost search (i.e., replace node expansion by GPT-3.5 as the world model).

We use the data from the Kaggle World cities database, select 68 cities with populations exceeding 5 million in different countries and 62 middle-size cities with populations between 500 thousand and 2 million, and use the Virtual Radar Server to get the flight routes dataset as ground truth. In our tests, we sampled 400 city pairs among large cities and 400 pairs among mid-size cities, evaluating path accuracy by verifying each direct flight exists. Paths were accepted if all flights were valid, even if they extended beyond our selected source and target cities. We evaluate the methods in two settings: predicting the flight route given two large cities and two mid-size cities.

**Results.** The main result shown in Fig 2 suggests that the LLM model + search algorithm consistently outperforms the LLM policy, supporting our analysis[2]. Furthermore, the performance gaps for mid-size cities are larger than for large cities. This is consistent with the fact that there are more mid-size cities than large cities (the gap between the description lengths of $n^2 \log n$ for policies vs $n \log n$ for models grows with the number of cities). Note that performance decreases as the path length increases for both methods. As the number of predictions required for each path increases with path length, the probability of incorrect path prediction also increases.

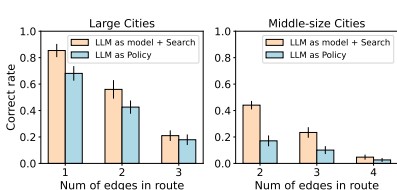

Figure 2: Flight planning results.

## 4.2 Object rearrangement task

Consider a house with $n$ movable objects, $m$ containers, and $k$ rooms. If we use L-Model, we must describe the model and search algorithm (i.e., MCTS). The model can be represented as a sparse graph with objects, containers, and rooms as nodes, and weighted directed edges signifying a "located in" relation between the nodes, with the weights specifying the probabilities. Assume that each weight is specified with a constant number of bits. Each of the $n$ objects can be located within the $m + k$ containers or rooms, so each object edge would require approximately $\log(m + k)$ bits to describe. Each of the $m$ containers can be located in $k$ rooms, so each container edge would require approximately $\log(k)$ bits to describe. Assume further that the degree of each object and container node in the graph is bounded by a constant; the entire graph then requires $O(n \log(m+k) + m \log(k)) = O((m + n) \log(m + k))$ bits to describe. The MCTS algorithm should be described by a programming language with a constant size. For the L-Policy, tasks can be designated using object-container pairs. Each object-container policy can be defined by a sequence of actions, e.g. "walk to the fridge, open the fridge," until the object is found, followed by a sequence of actions until the destination container is found. Each action takes $O(m + k)$ bits to describe. Assuming search sequences and the size of each object-container policy are bounded by a constant. Describing the policies for all $mn$ object-container pairs requires $O(mn \log(m + k))$ bits.

The composed tasks provide further challenges for L-Policy. Composing tasks increases the description complexity of policies to $O((mn)^N \log(m+k))$, where $N$ is the number of composed tasks if the composition is not exploited in the policies. For the L-Model, decomposition is automatically done in MCTS at the expense of more computation, whereas for the L-Policy, the LLM must learn to do the decomposition. This may make the problem of learning the decomposed policy computationally more difficult and less likely to be approximated by the LLM in practice.

This analysis indicates that the L-Model has a shorter description length than the L-Policy. According to the MDL principle, the L-Model will likely have a lower error rate than the L-Policy. However, in this case, we do not have an algorithm that is guaranteed to be efficient for solving L-Model. Instead, we use the L-Policy as a search heuristic to obtain a practically effective search algorithm. As predicted by the MDL principle, the search with the LLM-induced world model, when limited to the neighbourhood of the LLM-induced policy, provides improved performance over L-Policy.

## 4.3 Discussion

While we have discussed problems where the L-Model outperforms the L-Policy, we would also expect the L-Policy to outperform the L-Model when the description length of policies is shorter than

---

[2]Single-edge results are omitted for mid-size cities as there are no such routes in the experiments.

the models. For example, when recommending a tourist itinerary for a city, the description length of the itinerary should be shorter than describing all the places of interest plus the reward and the length of time recommended for visiting each location. In such a case, the LLM may be able to do better in recommending an itinerary than in providing accurate information on all places of interest for planning itineraries.

In the multiplication problem, the model-based approach used a known efficient multiplication algorithm. In the air travel planning task, an efficient shortest-path algorithm is used. However, for the object rearrangement problem, we do not have an efficient search algorithm. In this case, we demonstrate another strength of LLMs – as a policy, it can be used as a heuristic for improving the efficiency of search algorithms.

## 5 Related work

Task planning has a long-standing history in the AI research community. In the early stage, many studies [1, 15, 14, 12] focused on task planning in a discrete state space with deterministic transitions. These methods are intractable for large-scale, long-horizon problems. Recently, researchers have used learning-based methods to learn planning policies directly [22] or to learn search heuristics to accelerate planning [32, 31, 41, 7, 29]. Those policies or heuristics are not generalizable to other unseen settings. Most recently, the pre-trained LLMs have been applied as few-shot policies for task planning [18, 2, 19]. However, the planning policy may not have good compositional generalizablity. In the robotics community, many studies proposed that task planning should be integrated with physical-level motion planning, i.e., task and motion planning (TAMP) [11, 12, 9]. This paper focuses on large-scale task planning with partial observations and large, object-centric domains, in which classical planning is intractable and learning-based methods require massive data.

Various approaches are proposed to scale up to large-scale planning problems. Initially, the Monte Carlo method [21, 8, 6] is proposed to tackle intractable large-scale problems using random sampling in the tree search. However, further scaling up to the problem with a large domain with sparse rewards requires massive sampling. Silver et al. [32, 31] integrate MCTS with deep learning to bias the action selection and reduce sampling. The idea has been successfully applied in large-scale planning scenarios [32, 31, 7, 27]. Most recently, studies show that LLM can be a few-shot open-loop [18, 2] and closed-loop [19, 37] planning policy in the large domain, while it may suffer from hallucinations. In this paper, we show that LLMs' commonsense knowledge can guide a search algorithm, reducing the search space sufficiently for practical planning and producing reasoned results.

How to use LLMs for a planning problem? LLMs have been used as a few-shot policy for language-conditioned task planning [18, 2, 19], or a policy for multi-step reasoning problems [38, 42]. However, recent research [10] also suggests that transformer LLMs are inherently limited for solving multi-step reasoning problems. In addition, some studies try to use LLMs as heuristics [43] or transition function [13] in MCTS, boosting the performance in coding or small-scale reasoning. However, the literature has not discussed utilizing LLMs as a world model in depth. We show the benefits of using LLM to model the world, as well as using MDL analysis to decide how to use LLM for planning problems.

## 6 Conclusion

We use LLMs as the commonsense world model and the heuristic policy within MCTS to achieve better-reasoned decision-making for daily tasks. MCTS enables LLM to leverage its world modelling knowledge for informed reasoning and explore new combinations of actions to tackle novel tasks. LLM helps MCTS through the biased sampling of states and actions, improving its efficiency in resolving complex task-planning problems. Our analysis and empirical evidence suggest that, for certain real-world domains, if the description length of the world is substantially shorter than policy, using LLM as a model in a model-based approach is a better option than using LLM as policy.

**Limitations.** The runtime of LLM-MCTS is currently hindered by multiple LLM calls. The details of runtime performance are in Appendix E. While our method requires multiple LLM calls, it provides substantially improved results. There are also various ways to enhance runtime performance like using smaller LLMs like Llama [35, 36] or distilling LLM's knowledge into a smaller model [30, 16, 23]. Those are interesting avenues for future research.

**Broader impact.** There might be concerns about the inherent biases of LLMs that may lead to unfair or risky decisions in some domains. Further study about the fairness and bias of LLMs' knowledge would be beneficial.

## Acknowledgments and Disclosure of Funding

This research is supported in part by the National Research Foundation (NRF), Singapore and DSO National Laboratories under the AI Singapore Program (No. AISG2-RP-2020-016) and the Agency of Science, Technology and Research (A*STAR), Singapore, under the National Robotics Program (No. M23NBK0053).

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

# Appendix

## A Virtualhome experimental environments

We use the VirtualHome simulator [24] to evaluate our approach as well as the baseline methods. VirtualHome is a 3D household environment with partial observation, large action space, and a long planning horizon. It contains hundreds of interactive objects and containers, allowing it to perform various household object rearrangement tasks. This section introduces details of the tasks, the goal specifications, the actions, and the observations in our experimental settings.

### A.1 List of objects, containers, surfaces, and rooms in the apartment

We list all the objects that are included in our experimental environment. Here, we can put *moveable objects* into the *Containers* or on the *Surfaces*. The *Containers* and *Surfaces* are located at a *Room* in the apartment.

- *Containers*: bathroom cabinet, kitchen cabinet, bathroom counter, fridge, oven, dishwasher, microwave, stove, bathroom cabinet
- *Surfaces*: bed, bookshelf, cabinet, coffee table, cutting board, floor, fryingpan, kitchen counter, kitchen table, nightstand, sofa, stove
- *moveable objects*: alcohol, apple, banana, bar soap, bell pepper, boardgame, book, box, bread slice, bucket, candle, candy bar, carrot, cellphone, cereal, chicken, Chinese food, chips, chocolate syrup, clock, clothes pants, clothes pile, clothes shirt, coatrack, coffeepot, condiment bottle, condiment shaker, cooking pot, crackers, crayons, creamy buns, cupcake, cutlery fork, cutlery knife, cutlets, cutting board, dish bowl, dishwashing liquid, face cream, folder, fryingpan, glasses, globe, hair product, hanger, juice, keyboard, lime, lotion bottle, magazine, milk, milkshake, minced meat, mouse, mug, notes, oven tray, pancake, paper, pear, pie, pillow, plate, plum, poundcake, pudding, radio, remote control, salad, salmon, slippers, sports ball, sundae, teddybear, toilet paper, toothbrush, toothpaste, towel, towel rack, toy, washing sponge, water glass, whipped cream, wine, wineglass
- *Rooms*: bedroom, bathroom, living room, kitchen.

### A.2 Tasks

We use the object rearrangement tasks for evaluation. The task is to search for one or more objects in the house and move them to the desired positions. We use natural language as the interface to specify the tasks. Thus, the agent should take as input the natural language instruction and observations, and then output actions.

The tasks are randomly sampled from different distributions. We define various types of object rearrangement tasks for evaluation:

- *Simple*: this task is to move one object in the house to the desired location. The combination of the object and desired location has appeared in the training dataset.
- *Novel Simple*: this task is to move one object in the house to the desired location. The combination of the object and desired location has**not** appeared in the training dataset.
- *Comp.*: this task is composed of 2 *Simple* tasks, moving more than one object in the house to their desired location. This kind of task has a longer planning horizon as it requires moving multiple objects to complete. The combinations of *Simple* tasks have appeared in the training dataset.
- *Novel Comp. (2)*: this task is composed of 2 *Simple* tasks, moving more than one object in the house to their desired location. The combinations of *Simple* tasks have not appeared in the training dataset.
- *Novel Comp. (3)*: this task is composed of 3 *Simple* tasks, moving more than one object in the house to their desired location. This kind of task has the longest planning horizon. The combinations of *Simple* tasks have not appeared in the training dataset.

We also have different household environments:

- *Seen Apartment*: the map of the apartment is shown in Figure 3. These household environments are the same as the ones in the training set, while the object positions are randomly initialized according to a pre-defined commonsense distribution in VirtualHome [24].
- *Unseen Apartment*: the map of the apartment is shown in Figure 4. These household environments are not the same as the ones in the training set. The object positions are also sampled from a different pre-defined commonsense distribution in VirtualHome [24].

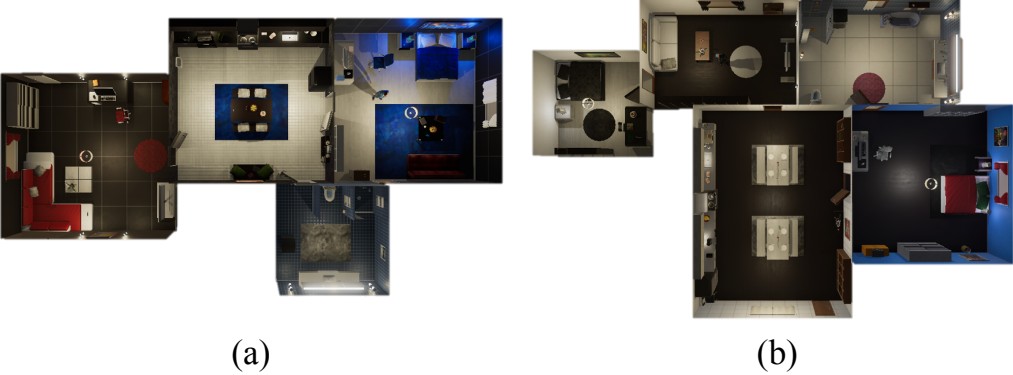

(a)      (b)

Figure 3: The map of the *seen apartments* in our setting. These household environments are the same as the ones in the training set, while the object positions are randomly initialized according to a commonsense distribution.

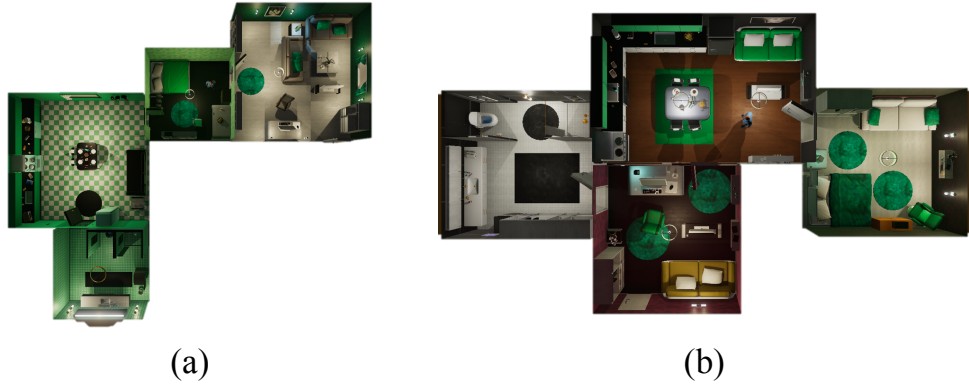

(a)      (b)

Figure 4: The map of the *unseen apartments* in our setting. These household environments are not the same as the ones in the training set. The object positions are also sampled from a different commonsense distribution.

### A.3   Goal specification

Similar to prior works [22], we define the goal in the VirtualHome system by a set of predicates. For instance, a goal can be defined by *Inside(apple, fridge):2; Inside(plate, dishwasher):1*, meaning "put two apples inside the fridge and put one plate inside the dishwasher." For *Simple* and *Novel Simple* tasks, it only requires moving one object, while *Comp.* and *Novel Comp.* have more than one object to move.

### A.4   Actions

In VirtualHome, the agent is able to navigate in the environment, grab an object, put an object inside the containers (e.g., fridge) or on the surfaces (e.g., table), open and close the container, etc. The

actions in VirtualHome are grounded to moveable objects, containers, or rooms in the environment. For example, `Open(5)` is to open an object with index (5). The list of available actions in our setting are listed below:

- `Walk(<item>)`: walk to the `<item>`. The `<item>` can be a moveable object, a container, or a room. The precondition of this action is that the `<item>` is visible. The effect of this action is that the agent is close to the `<item>` if the `<item>` is an object or inside the `<item>` if the `<item>` is a room. The action is translated into the sentence "walk to the `<name of item>`" when feeding into LLMs.
- `Open(<item>)`: open the `<item>`. The `<item>` can be a moveable object or a container. The precondition of this action is that the agent should be close to `<item>`. The effect of this action is that the `<item>` is opened. The action is translated into the sentence "open the `<name of item>`" when feeding into LLMs.
- `Close(<item>)`: close the `<item>`. The `<item>` can be a moveable object or a container. The precondition of this action is that the agent should be close to `<item>`. The effect of this action is that the `<item>` is closed. The action is translated into the sentence "close the `<name of item>`" when feeding into LLMs.
- `Grab(<item>)`: grab the `<item>`. The `<item>` should be a moveable object. The precondition of this action is that the agent should be close to the `<item>`, and the agent is not holding any objects. The effect of this action is that the agent will hold the `<item>`. The action is translated into the sentence "grab the `<name of item>`" when feeding into LLMs.
- `PutIn(<item1>, <item2>)`: put the moveable object `<item1>` inside the container `<item2>`. The precondition of this action is that the agent should be close to the `<item2>` and holding `<item1>`. The effect of this action is that the agent is not holding any objects, and the `<item1>` is inside the `<item2>`. The action is translated into the sentence "put the `<name of item1>` inside the `<name of item2>`" when feeding into LLMs.
- `PutBack(<item1>, <item2>)`: put the moveable object `<item1>` on the surface `<item2>`. The precondition of this action is that the agent should be close to the `<item2>` and holding `<item1>`. The effect of this action is that the agent is not holding any objects, and the `<item1>` is on the `<item2>`. The action is translated into the sentence "put the `<name of item1>` on the `<name of item2>`" when feeding into LLMs.

### A.5 Observations

We use the same representation as [22] for partial observation. The observation is a list of visible objects and relationships between those objects. Each object or container has a state: *open* or *close*. The fine-tuned GPT2 policy [22] also uses the 3d coordinates of the object. We also use relationships to connect different objects, such as `Inside(apple, fridge)`. Those relationships are translated to natural language descriptions when feeding into LLMs, such as "an apple is inside the fridge."

## B  Data gathering

Similar to prior works [25, 22], we collect expert trajectories in VirtualHome using regression planning with handcrafted heuristics[3]. The expert has full observation of the environment. Given the goal predicates and full observation, the agent will use the handcrafted heuristics for each task to effectively search for the solutions. The expert also has a handcrafted mechanism for compositional tasks to decompose one task into subtasks and finish them progressively. For each trajectory, we include the goal predicates (used by the VirtualHome system and the expert agent), the goal instruction (used by the agent), the partial observation for each time step (not used by the expert agent, the expert agent uses full observation), and the expert actions.

## C  Implementation details of belief in LLM-MCTS

This section introduces our implementation details for the belief of states in GPT3.5-MCTS. The source code will be released at https://llm-mcts.github.io before the publication.

---

[3]Their implementation is available at: https://github.com/xavierpuigf/watch_and_help.git

### C.1 State representation

We represent the states by a list of objects and their relationships. Each object has a unique name and id in the simulator, as well as the state of the object. We use the same unique name and id in our state representation. The relationships connect different objects, containers, surfaces, and rooms. The VirtualHome contains 59 different types of relationships, including `Inside, On, Close, Facing`, etc. We use the same type of relationships in our state representation.

### C.2 Belief

The belief of the state also contains a list of objects and their relationships. However, we parameterize the relationships by a vector, representing the probability that the relationship is true. This vector is affiliated with the object representation. For simplicity, we only include the relationships `Inside`, `On` in our belief, as we only query LLM about the object positions to build up the commonsense belief of the state.

When building up a state's belief, we query LLM to predict the position of each moveable object, container, and surface. The position of a moveable object is specified by the relationships (i.e., `Inside` or `On`) between itself and a container or surface. The position of a container or a surface is specified by its relationship (i.e., `Inside`) to the room. We use sampling to approximate the distribution of the position. The moveable objects' belief of position is represented by a vector whose dimension is the same as the total number of containers and surfaces in the house. Each vector entry denotes the probability that whether the object is inside a specific container or on a specific surface is true. When asking LLM to predict the object positions, we asked LLM for $M$ times and received multiple responses from LLM. We then count each entry's total number of predictions and normalize them to become a probability distribution. We initialize the value of other unsampled entries in the vector by a lower bound of the probability $1 \times 10^{-3}$ to ensure that the model will not eliminate other possibilities when the commonsense model is wrong.

The agent will receive new observations to update their belief when interacting with the environment. We will first predict the next state of the agent by the transition function and then update the belief of the object positions by new observations. Suppose the object is inside the current observation. In that case, the other entry of the relations between objects will be masked out by zero, and the entry of the relationships in observation will be replaced by the value of one. However, if a relationship is not inside the observation, the value of the corresponding entry will be replaced by zero, and the vector will be normalized again.

## D   Visualized examples

We provide a set of successful (shown in Figure 5) and failed trajectories (shown in Figure 6) to give a better understanding of the tasks and our method. Policy, model, and translation errors are the primary causes of failures. Among these, policy errors are responsible for the majority of the failures. Often time, the policy produces unreasonable behaviors that mislead the search procedure. For example, it usually outputs inadmissible actions, such as "*walk to the cutlery fork*" where the "*cutlery fork*" is not in the observation (shown in Figure 6 (a)). It also produces back-and-forth behaviors, resulting in an unreasonable heuristic and slowing the search procedure. For example, when putting objects inside the microwave, it is sometimes struck by repeatedly opening and closing the microwave. For model error, the predicted positions of objects are not always correct. Since a random rollout policy is employed, incorrect object states can result in higher $Q$-values than correct states, leading to misguided exploration (shown in Figure 6 (b)). The wrong translation also compromises the performance as we translate the response from LLM to admissible action or object names to ensure executability. This is caused in part by the VirtualHome environments, as the policy might not understand the underlying logic of the actions in VirtualHome, such as you have to walk close to interact with the object. Thus, if the LLM outputs "open fridge" but is not close enough to the fridge, the action will be translated to other admissible actions ("open fridge" is not inside the admissible actions for this case as it is invalid due to the setting of VirtualHome).

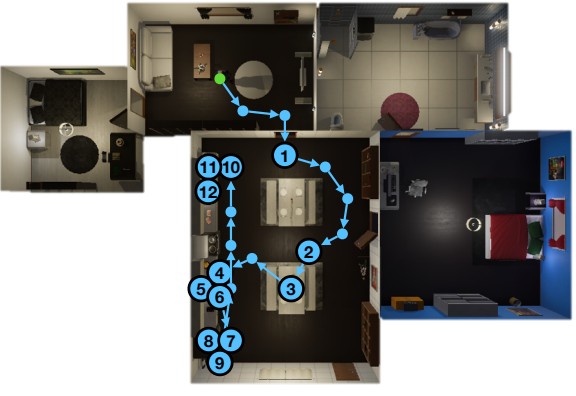

Instruction: Put one plate inside the kitchen cabinet and one chicken inside microwave.
1: Walk to kitchen
2: Walk to plate
3: Grab plate
4: Walk to kitchen cabinet
5: Open kitchen cabinet
6: Put plate inside kitchen cabinet
7: Move fridge
8: Open fridge
9: Grab chicken
10: Walk to microwave
11: Open microwave
12: Put chicken inside microwave

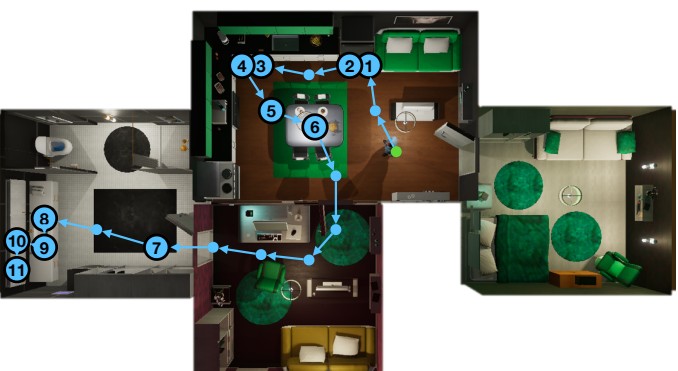

Instruction: Put one apple on the kitchen table and one toothbrush inside the bathroom cabinet.
1: Walk to fridge
2: Open fridge
3: Walk to apple
4: Grab apple
5: Walk to kitchen table
6: Put apple on kitchen table
7: Walk to bathroom
8: Walk to toothbrush
9: Grab toothbrush
10: Open bathroom cabinet
11: Put toothbrush inside bathroom cabinet

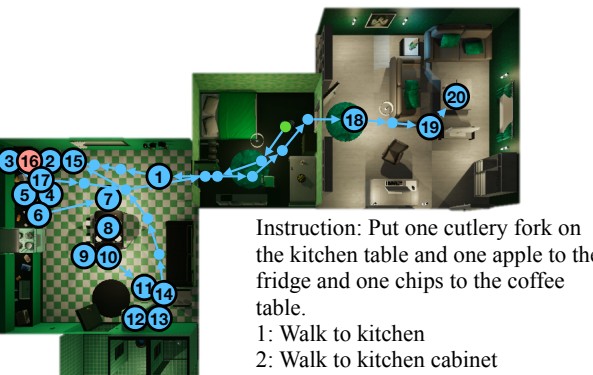

Instruction: Put one cutlery fork on the kitchen table and one apple to the fridge and one chips to the coffee table.
1: Walk to kitchen
2: Walk to kitchen cabinet
3: Open kitchen cabinet
4: Walk to dishwasher
5: Open dishwasher
6: Grab cutlery fork

7: Walk to kitchen table
8: Put cutlery fork on kitchen table
9: Walk to apple
10: Grab apple
11: Walk fridge
12: Open fridge
13: Put apple inside fridge
14: Close fridge
15: Walk to kitchen cabinet
16: Open kitchen cabinet (Failed)
17: Grab chips
18: Walk to living room
19: Walk to coffee table
20: Put chips on coffee table

Figure 5: Successful examples

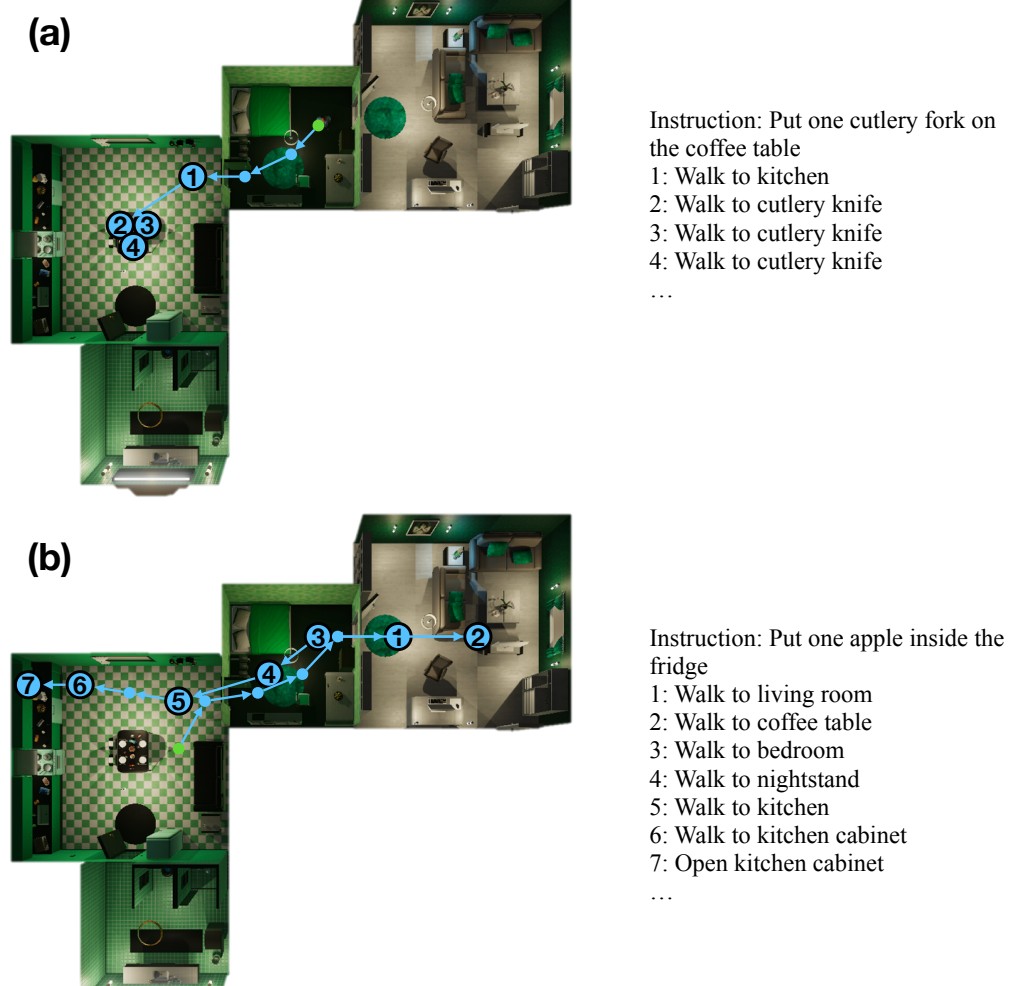

**(a)**

Instruction: Put one cutlery fork on the coffee table
1: Walk to kitchen
2: Walk to cutlery knife
3: Walk to cutlery knife
4: Walk to cutlery knife
…

**(b)**

Instruction: Put one apple inside the fridge
1: Walk to living room
2: Walk to coffee table
3: Walk to bedroom
4: Walk to nightstand
5: Walk to kitchen
6: Walk to kitchen cabinet
7: Open kitchen cabinet
…

Figure 6: Failed examples. (a) Policy error and translation error. LLM outputs walk to the cutlery fork, but the cutlery fork is not in observation. We use embeddings to evaluate the most similar valid actions. Therefore it translates the action to one similar action "walk to cutlery knife." The action has an incorrect semantic meaning and causes failure. (b) model error. The LLM predicts the apple is on the nightstand in the bedroom and on the coffee table in the living room. As we are using random rollout to get the estimation of the reward, there will be situations when the incorrect actions result in a higher estimated $Q$ value, thereby misleading the exploration.

Table 3: Runtime performance (second $\pm$ standard error)

| | GPT3.5-MCTS | GPT3.5 Policy | UCT | GPT2 Policy |
|---|---|---|---|---|
| Runtime | $67.83 \pm 18.92$ | $1.19 \pm 0.81$ | $120.0 \pm 0.0$ | $0.33 \pm 0.07$ |

## E  Runtime performance

The runtime performance to make one-step decisions for simple object-rearrangement tasks is reported in Fig E. The experimental setup is the same as our main experiments in VirtualHome. The simple tasks are the tasks that only require the rearrangement of one item generated from the same distribution as the training dataset. We used 100 times simulation during the tree search for GPT3.5-MCTS in our experiments. For UCT, we bound the runtime by 120 seconds. The details of our hardware for experiments are enclosed below:

- CPU: Intel(R) Xeon(R) Gold 6240 CPU @ 2.60GHz (72 cores)
- GPU: NVIDIA GeForce RTX 2080 Ti

## F  Prompts

One example prompt for the LLM policy and the exact prompt we used for building up the commonsense belief is shown below.

Listing 1: Example prompt for the heuristic policy

```
1 You need to generate a high-level plan for completing a household task using the allowed actions and
       visible objects.
2 Allowed actions: walk to <object>, walk to <room>, walk to <container>, walk to <surface>, grab <object>,
       open <container>, close <container>, put <object> on <surface>, put <object> inside <container>.
3 Rooms in the house: bedroom, bathroom, living room, kitchen
4 You need to strictly follow the format in the following examples:
5 Goal: Put one apple inside the fridge
6 Completed actions: walk to the kitchen, walk to the apple
7 Current Observation: a kitchen table is inside the kitchen, a kitchen counter is inside the kitchen, an
       apple is on the kitchen counter, a plate is on the kitchen table, a banana is on the kitchen counter,
        a fridge is inside the kitchen and fridge is closed, a kitchen cabinet is inside the kitchen and
       kitchen cabinet is closed, a cutlery knife is on the kitchen table, a microwave is inside the kitchen
        and microwave is closed, a dishwasher is inside the kitchen and dishwasher is closed.
8 Next actions: grab the apple, walk to the fridge, open the fridge, put the apple inside the fridge, close
       the fridge, done.
9 Now, finish the next following task.
10
11 Goal: Put one apple on the kitchen table
12 Completed actions: walk to the kitchen
13 Current observation: a kitchen table is inside the kitchen, an apple is on the kitchen table, a kitchen
       counter is inside the kitchen, an apple is on the kitchen counter, a cutlery knife is on the kitchen
       counter, a fridge is inside the kitchen and fridge is closed, a kitchen cabinet is inside the kitchen
        and kitchen cabinet is closed, a kitchen table is inside the kitchen, a plate is on the kitchen
       table, a pounding cake is on the kitchen table, a microwave is inside the kitchen and microwave is
       closed, a dishwasher is inside the kitchen and dishwasher is closed.
14 Next actions:
```

Listing 2: Example prompt for the commonsense world model

```
1 You need to predict the positions of the moveable objects, containers, and surfaces in the apartment
       according to the commonsense.
2 Rooms in the apartment: bedroom, bathroom, living room, kitchen.
3 Containers in the apartment: bathroom cabinet, kitchen cabinet, bathroom counter, fridge, oven, dishwasher,
        microwave, stove, bathroom cabinet.
4 Surfaces in the apartment: bed, bookshelf, cabinet, coffee table, cutting board, floor, fryingpan, kitchen
        counter, kitchen table, nightstand, sofa, stove.
5 You need to strictly follow the format in the following examples:
6 Question: what are the possible positions of strawberry?
7 Answer: Inside fridge, On kitchen table.
8 Question: what are the possible positions of soap?
9 Answer: On bathroom counter.
10 Question: what are the possible positions of water cup?
11 Answer: On kitchen table, Inside dishwasher.
12 Now, answer the next following question.
13 Question: what are the possible positions of apple?
```

```
14  Answer:
```

Listing 3: Example prompt for the natural language instruction interpretation

```
1 You need to interpret the natural language goal into a formal goal representation
2 For example,
3 Goal: put 1 toothbrush inside the bathroomcabinet.
4 Formal goal: (INSIDE, toothbrush, bathroomcabinet, 1)
5 Goal: put 1 toothbrush inside the bathroomcabinet, put 1 apple on the kitchentable.
6 Formal goal: (INSIDE, toothbrush, bathroomcabinet, 1)-(ON, apple, kitchentable, 1)
7 Goal: put 1 toothbrush inside the bathroomcabinet, put 1 apple on the kitchentable, put 1 chicken inside
         the fridge.
8 Formal goal: (INSIDE, toothbrush, bathroomcabinet, 1)-(ON, apple, kitchentable, 1)-(INSIDE, chicken,
         fridge, 1)
9 Now, interpret the next following goals:
```

