# OpenReview forum: "Large Language Models as Commonsense Knowledge for Large-Scale Task Planning"
_NeurIPS.cc/2023/Conference — NeurIPS 2023 poster_

### Official Review · Reviewer_euNw · 2023-06-07

**Soundness:** 3 good
**Presentation:** 3 good
**Contribution:** 4 excellent
**Rating:** 7
**Confidence:** 4

**Summary:**

The paper proposes to use large language models (LLMs) as a world model instead of a policy for task planning. Specifically, the LLM is used to approximate the state of the world and acts as a heuristic policy in Monte-Carlo Tree Search (MCTS). Experimental results demonstrate the effectiveness of the method, outperforming a finetuned small model and a LLM policy.

**Strengths:**

1. The paper innovatively proposes to use language model as world model instead of policy model for task planning. The idea is quite interesting and the motivation is convincing, i.e., this can lower the complexity of the problem and better utilize the encoded commonsense knowledge in language models.
2. The experimental results demonstrates the effectiveness of the proposed method.

**Weaknesses:**

1. It is better to include more baselines that employ special design for task planning, e.g., SayCan [1], Zero-Shot Planner [2], etc.
2. Only simple settings are considered in this work, i.e., object re-arranging task in a household environment. I would be interested in seeing if the method will still work when applied to more complex environments, which contain more diverse objects making it harder to predict world state.



[1] Ahn et al. Do As I Can, Not As I Say: Grounding Language in Robotic Affordances
[2] Huang et al. Language Models as Zero-Shot Planners: Extracting Actionable Knowledge for Embodied Agents

**Questions:**

LLM-MCTS retrieves in-context examplars from a dataset. Does the baseline model also do this?

**Limitations:**

Seems that I don't see a limitation section in the paper? Also, broader societal impacts are not included, which should typically be consider for a generation model like large language models. Please refer to weakness section for improvement.

---

> ### Author Rebuttal · Authors · 2023-08-09
>
> Thank you very much for your valuable feedback. We will improve and revise the paper according to your suggestions. Our reply to your question is enclosed below.
>
> Q1:
> > It is better to include more baselines that employ special design for task planning, e.g., SayCan [1], Zero-Shot Planner [2], etc.
>
> A1: The key idea of SayCan is to learn physical-level affordance from raw observation when executing physical actions. In high-level task planning, the affordance is determined by the precondition of the pre-defined actions. We are keen to compare with SayCan when combining our method with physical-level action policy. Our GPT-3.5 policy baseline is essentially [2] with a one-shot example prompt and observation feedback, as we want to provide the same information for a fair comparison.
>
> Q2:
> > Only simple settings are considered in this work, i.e., object re-arranging task in a household environment. I would be interested in seeing if the method will still work when applied to more complex environments, which contain more diverse objects making it harder to predict world state.
>
> A2: Object rearrangement is a representative embodied AI task [3-9] with many practical implications in everyday life, such as setting the table, tidying up the room, loading the dishwasher, and more. Thus, object rearrangement experiments are an interesting setting to investigate a fairly large set of planning capabilities required in embodied AI. We chose VirtualHome, as it is an established domain well used in prior work [2,13,14]. Our method is domain-agnostic and should be able to generalize to other common household domains, as LLMs have vast general knowledge that should be widely applicable [10-12]. We will explore those distinct datasets for further evaluation.
>
> Q3:
> > LLM-MCTS retrieves in-context exemplars from a dataset. Does the baseline model also do this?
>
> A3: Yes, the baseline GPT-3.5 policy has the same mechanism as the heuristic policy in GPT3.5-MCTS. GPT2 policy, however, is fine-tuned by the entire training dataset using behavior cloning.
>
> Q4:
> > Seems that I don't see a limitation section in the paper? Also, broader societal impacts are not included, which should typically be consider for a generation model like large language models.
>
> A4: Due to the page limit, we briefly introduce the limitation in the failure analysis and conclusion. We will consider the potential broader impact and add it to the manuscript during revision.
>
> [1] Ahn et al. Do As I Can, Not As I Say: Grounding Language in Robotic Affordances
>
> [2] Huang et al. Language Models as Zero-Shot Planners: Extracting Actionable Knowledge for Embodied Agents
>
> [3] B. Dhruv et al., “Rearrangement: A Challenge for Embodied AI.” 2020.
>
> [4] L. Weihs et al. "Visual room rearrangement." CVPR 2021.
>
> [5] A. Szot et al. "Habitat 2.0: Training home assistants to rearrange their habitat." NeurIPS 2021.
>
> [6] Y. Kant et al. "Housekeep: Tidying virtual households using commonsense reasoning." ECCV 2022.
>
> [7] A. Khandelwal et al. "Simple but effective: Clip embeddings for embodied ai." CVPR 2022.
>
> [8] E. Huang et al.,  "Large-scale multi-object rearrangement." ICRA 2019.
>
> [9] A. Krontiris et al. "Dealing with Difficult Instances of Object Rearrangement." RSS 2015.
>
> [10] S. Bubeck et al. "Sparks of artificial general intelligence: Early experiments with gpt-4." arXiv preprint arXiv:2303.12712 (2023).
>
> [11] T. Silver et al. Generalized Planning in PDDL Domains with Pretrained Large Language Models. arXiv preprint 2023.
>
> [12] B. Liu et al. "Llm+ p: Empowering large language models with optimal planning proficiency." arXiv preprint arXiv:2304.11477 (2023).
>
> [13] I. Singh et al., “ProgPrompt: Generating Situated Robot Task Plans using Large Language Models”, ICRA 2023.
>
> [14] S. Li et al., “Pre-trained language models for interactive decision-making,” Neurips 2022.

---

### Official Review · Reviewer_Mf2f · 2023-07-03

**Soundness:** 3 good
**Presentation:** 2 fair
**Contribution:** 3 good
**Rating:** 7
**Confidence:** 4

**Summary:**

This paper proposed to leverage LLMs both as a (commonsense) world model and heuristic policy within the MCTS search algorithm to tackle household planning tasks namely object rearrangements. The main idea is that for each simulation phase in MCTS, the algorithm sample from LLM to obtain the initial belief of states (of objects) and then use the LLM as a heuristic policy to guide action selection and finding promising trajectories.
[Some details of when and how often the LLM is used as a world model is missing from the main paper (see questions)]

The paper evaluated their approach using a subset of VirtualHome tasks– object rearrangement. They tested models on simple and compositional (rearranging multiple objects) tasks, as well as in-distribution and out-of-distribution settings. They use the `success rate’ of completing the tasks within 30 steps as evaluation metrics for comparison. They demonstrate significant improvements over baselines including a variant of the MCTS wo/ commonsense world model, a supervised GPT-2 model, and using GPT-3.5 only as the action policy. The improvements are larger for compositional and OOD setups where their method benefits from the MCTSC’s lookahead and LLM’s commonsense knowledge of the world.


**Strengths:**

- Interesting/timely approach to leveraging LLM both as a world model and policy within the MCTS framework for the important task of planning
- Significant improvements over baselines
- Thorough and insightful ablation study to analyze the functioning of different components


**Weaknesses:**

- The paper only tested on object rearrangement task with limited object relationships (on, inside). More complex and realistic tasks are left unexplored.
- The paper benefits from the rewriting of Sec 4.2 to add technical details. At its current state, it’s unclear how often the LLM is used as a world model (see questions).


**Questions:**

1– From my understanding the LLM is only used once (per simulation) to get the initial state of the objects. How do you use that to estimate the value of selected actions (line 175)? Do you get different world states for different simulation iterations? is the commonsense model used at any other stage in the MCTS search algo? I think a more organized description of the process with a running example would be helpful.


2- During MCTS, to sample state from the belief, do you sample the position of all available objects or task-related objects?

3- Some notations are undefined in Algorithm 1: d, tau, d’, …

4- Why did the author limit themselves to only object rearrangement tasks with only a few object relationships (on, inside)? Did the author explore their method on other household or planning tasks?

5- For compositional tasks, are models provided with 1 few-shot compositional example or a simple example?

6- Line 291: how is the ground-truth reward function obtained? It's unclear how the ground-truth reward function is different from the one used in the proposed final model

7- An interesting observation is that sometimes GPT3.5 and MCTS outperform in the unseen apartment setup compared to the seen one (on some tasks). Do authors have any insight/speculation on this?


**Limitations:**

Yes, briefly in conclusion.

---

> ### Author Rebuttal · Authors · 2023-08-09
>
> Thank you very much for your valuable feedback. We will carefully consider and incorporate your comments and suggestions into our manuscript. The reply to the questions is enclosed below.
>
> Q1:
> > The paper only tested on object rearrangement tasks with limited object relationships (on, inside). More complex and realistic tasks are left unexplored.
>
> A1: While our experiments focus on object rearrangement, we believe that it is appropriate as object rearrangement is a representative embodied AI task [1-7] with practical implications in everyday life, such as setting the table, tidying up the room, loading the dishwasher, and more. Thus, we believe that object rearrangement experiments reasonably support our claims in the paper.
>
> Object relationships (on, inside) are only used to initialize the belief of the world. Using an imperfect but reasonable world model and object positioning is a trade-off between efficiency and accuracy. As the agent navigates and receives new observations, its beliefs of states and relationships of objects will be updated.
>
> Q2:
> > The paper benefits from the rewriting of Sec 4.2 to add technical details. At its current state, it’s unclear how often the LLM is used as a world model (see questions).
>
> A2: We will revise Sec 4.2 according to your questions. Please see our responses below.
>
> Q3:
> > How do you use that to estimate the value of selected actions (line 175)? Do you get different world states for different simulation iterations?
>
> A3: In MCTS, we sample a world for each simulation, and the sampled world could be different for each simulation. In one simulation, the agent selects actions and sample observations according to the world in the root until it reaches the leaf node of the tree for expansion and rollout. It results in a trajectory of the tree with a reward. The root will sample a different world in different simulations, resulting in different trajectories in the same tree. We back up all the rewards from all the obtained trajectories to get the approximated Q-function at the root (Alg1, line 34 recursively updates to the root). The original paper on MCTS for the POMDP [11] may provide additional understanding.
>
> > is the commonsense model used at any other stage in the MCTS search algo?
>
> The LLM is used not only in sampling possible worlds but also as a search heuristic (in Alg1, line 29, LLM is used as heuristic policy: $\hat{\pi}(a|h)$).
>
> Q4:
> > do you sample the position of all available objects or task-related objects?
>
> A4: We sample all available objects of the world.
>
> Q5:
> > Some notations are undefined in Algorithm 1: d, tau, d’, …
>
> A5: $d$ and $d’$ is the current depth of the tree, and $\mathcal{T}$ denotes the current tree. We will explain the notations in future revisions.
>
> Q6:
> > Why did the author limit themselves to only object rearrangement tasks with only a few object relationships (on, inside)? Did the author explore their method on other household or planning tasks?
>
> A6: See our response A1.
>
> Q7:
> > For compositional tasks, are models provided with 1 few-shot compositional example or a simple example?
>
> A7: The semantic similarity determines the selection of examples. If the cosine similarity between the embeddings of current instruction and instruction in the dataset is higher than the others, the instruction in the dataset and its corresponding trajectory are then selected as the example. The dataset contains examples of compositional tasks; thus, the compositional example will likely be selected if the current task is compositional.
>
> Q8:
> > Line 291: how is the ground-truth reward function obtained? It's unclear how the ground-truth reward function is different from the one used in the proposed final model
>
> A8: We assume that the baseline UCT method has the ground-truth reward function. Otherwise, it will not be able to plan. For each task, we have the ground-truth goal, which we use to determine the reward in the UCT baseline. The proposed method uses LLM to interpret the natural language instructions into the goal state, and the goal state determines the reward function, i.e., there will be a positive reward if the goal is achieved.
>
> Q9:
> > An interesting observation is that sometimes GPT3.5 and MCTS outperform in the unseen apartment setup compared to the seen one (on some tasks). Do authors have any insight/speculation on this?
>
> A9: Some unseen domains are not as large as the example domains. There might be some variance due to the uncertainty of sampling in both the GPT3.5 and MCTS. Occasionally, it has a higher success rate when the actual performance difference is small.
>
> [1] B. Dhruv et al., “Rearrangement: A Challenge for Embodied AI.” 2020.
>
> [2] L. Weihs et al. "Visual room rearrangement." CVPR 2021.
>
> [3] A. Szot et al. "Habitat 2.0: Training home assistants to rearrange their habitat." NeurIPS 2021.
>
> [4] Y. Kant et al. "Housekeep: Tidying virtual households using commonsense reasoning." ECCV 2022.
>
> [5] A. Khandelwal et al. "Simple but effective: Clip embeddings for embodied ai." CVPR 2022.
>
> [6] E. Huang et al.,  "Large-scale multi-object rearrangement." ICRA 2019.
>
> [7] A. Krontiris et al. "Dealing with Difficult Instances of Object Rearrangement." RSS 2015.
>
> [8] S. Bubeck et al. "Sparks of artificial general intelligence: Early experiments with gpt-4." 2023.
>
> [9] T. Silver et al. Generalized Planning in PDDL Domains with Pretrained Large Language Models. 2023.
>
> [10] B. Liu et al. Llm+ p: Empowering large language models with optimal planning proficiency. 2023.
>
> [11] D. Silver et al. Monte-Carlo planning in large POMDPs. NeurIPS, 2010.
>
> [12] I. Singh et al. ProgPrompt: Generating Situated Robot Task Plans using Large Language Models. ICRA 2023.
>
> [13] S. Li et al. Pre-trained language models for interactive decision-making. Neurips 2022.
>
> [14] W. Huang et al. Language Models as Zero-Shot Planners: Extracting Actionable Knowledge for Embodied Agents, ICML 2022.

---

> > ### Comment · Reviewer_Mf2f · 2023-08-13
> > **Thank you for your response**
> >
> > Thank you for your detailed responses to clarify details. I believe including these in the paper will greatly improve the readability of the paper.
> >
> > Despite some missing details which their inclusion will greatly improve readability, this is a solid paper and I am leaning towards positive recommendation.

---

> > > ### Author Response · Authors · 2023-08-13
> > >
> > > Thank you for offering your valuable suggestions! They have helped us significantly in improving the manuscript.

---

### Official Review · Reviewer_RKS5 · 2023-07-05

**Soundness:** 2 fair
**Presentation:** 2 fair
**Contribution:** 2 fair
**Rating:** 4
**Confidence:** 4

**Summary:**

The paper introduces a new methodology _Monte Carlo planning with common sense knowledge_. The idea is to rely on LLMs to integrate common background knowledge into Monte Carlo Tree Search algorithm with application to language-instructed object rearrangement tasks.

Assuming access to a dataset of expert actions and observations, a list of all objects, containers and surfaces appearing in the dataset are retrieved. Similar to [S.Li 2022](https://arxiv.org/pdf/2202.01771.pdf) an LLM is used to approximate the belief of the state, containing the list of objects and their relationship. For instance, the fridge is likely to be in the kitchen.

To derive a policy, an LLM is also used, building on the work of [S.Li 2022](https://arxiv.org/pdf/2202.01771.pdf). Relying on PUCT, the model takes as input the examples in the dataset, the goal description, the current observations and the history of actions and outputs the list of next actions to take.

The method developed is evaluated on the Virtual Home, where the task is to rearrange objects in different apartments. The complexity of the task depends on the novelty of the apartment and the object considered (could be observed or not in the training data). The benchmark includes four methodologies, UCT, two baselines based on [S.Li 2022](https://arxiv.org/pdf/2202.01771.pdf) and the proposed methodology. The results demonstrate the improvement induced by _Monte Carlo planning with commonsense knowledge_ independently of the complexity of the task. An ablation study demonstrates the improvement induced by the initial approximation of the belief of the state. Finally an analysis of failures is conducted and demonstrates that most of them are linked to inadmissible actions outputted by the LLMs and back-and-forth behavior.

**Strengths:**

The work builds on [S.Li 2022](https://arxiv.org/pdf/2202.01771.pdf), relying on an LLM to approximate the belief of the state and to derive a policy. The novelty is to rely on MCTS and PUCT to derive the policy instead of DT. The approach is evaluated on a broader task than [S.Li 2022](https://arxiv.org/pdf/2202.01771.pdf), including compositional task that are supposed to be more complex than simple task to solve.

Finally, the ablation study provides insights on the added value of each block of the proposed methodology.

**Weaknesses:**

The paper mostly relies on experiments and ideas from [S.Li 2022](https://arxiv.org/pdf/2202.01771.pdf) and would gain clarity if clearly stated. For instance, the use of LLMs to initialize the belief state was already used in [S.Li 2022](https://arxiv.org/pdf/2202.01771.pdf).

Additionally, the code is not public and the implementation of LLM_MCTS is not super clear form pseudo code available in _Algorithm 1_. The paper would gain clarity by making the code public or available in supplementary materials.

**Questions:**

The parameters $c_{puct}$ and $c_{uct}$ can have a huge impact on the performance of the algorithm. How did you select the parameters?

PUCT can struggle with large state spaces, did it motivate the choice to keep only 2 types of relationship instead of 59 proposed in the task. Getting insights on the potential drop in performance induced by an increase in the size of the state space would be interesting.

200 expert trajectories are used in the training set. Having an idea of the impact of the number of trajectories on the performance of the methodology could be of great interest.

**Limitations:**

The two main limitations of the methodology are the non admissible outputs produced by the model and the back and forth behavior. The two were acknowledged by the authors.

---

> ### Author Rebuttal · Authors · 2023-08-09
>
> We sincerely appreciate your effort in reviewing our paper and providing feedback. Please see our responses below.
>
> Q1:
> > The paper mostly relies on experiments and ideas from S.Li 2022. and would gain clarity if clearly stated. For instance, using LLMs to initialize the belief state was already used in S.Li 2022.
>
> A1: We believe that the reviewer may have missed out on our paper's main contribution and misunderstood the method of Li et al. [2]:
> * While our research mainly contributes to using GPT-3.5 as a commonsense world model for **model-based search** with MCTS, Li et al. [2] use behavior cloning to finetune GPT-2 as a **model-free policy**. There are fundamental differences between the model-based and the model-free approaches, conceptually and algorithmically. In particular, the power of the model-based approach lies in its ability to compose elementary pieces of (commonsense) world knowledge through a reasoning/planning procedure (MCTS, in our case) and achieve compositional generalization. This is often one key weakness of the model-free approach.
> * Li et al.'s approach [2], by its nature, would not employ belief states in planning, contradicting RKS5's assertion.
> * Li et al. [2] only mentioned belief states in expert data collection (Appendix E.1.). It clearly states that they use the code of [1] for implementation, in which the belief of the initial state is a uniform distribution rather than initialized by LLM.
>
> We hope that our response can clarify the misunderstanding.
>
> Q2:
> > To derive a policy, an LLM is also used, building on the work of S.Li 2022.
>
> A2: Our heuristic policy uses LLM but is not built on Li et al. [2] as we do not finetune the LLM using behavior cloning.
>
> Q3:
> > The benchmark includes…, two baselines based on S.Li 2022 ...
>
> A3: We only use one baseline from Li et al [2], i.e., finetuned GPT-2 policy using behavior cloning. Other baselines are UCT and the few-shot GPT3.5 policy adapted from [3]. Please take a look at [3], as it is substantially different from [2] as it does not fine-tune the LLM, but uses prompts to conduct few-shot/zero-shot planning.
>
> Q4:
> > Additionally, the code is not public and the implementation of LLM_MCTS is not super clear form pseudo code available in Algorithm 1. The paper would gain clarity by making the code public or available in supplementary materials.
>
> A4: We intend to make the code publicly available for the camera-ready version of the paper. In the meantime, we provide some implementation details here. Our dataset generation is adapted from the code in the paper [1]. The fine-tuned GPT2 and few-shot GPT3.5 policies are from [2] and [3]. Our UCT and LLM-MCTS implementations are adapted from [4].
>
> Q5:
> > The parameters $c_{puct}$ and $c_{uct}$ can have a huge impact on the performance of the algorithm. How did you select the parameters?
>
> A5: We do limited tuning of the parameters within a range, initially testing UCT and LLM-MCTS in a small domain to verify correctness. After efficiently solving the problem in this small domain, we applied the method to a larger domain, increasing and continually tuning parameters to balance the exploration and exploitation. Lack of access to OpenAI GPT-3.5 and time constraints hindered more advanced tuning methods like Bayesian Optimization, leaving potential room for performance improvement.
>
> Q6:
> > PUCT can struggle with large state spaces, did it motivate the choice to keep only 2 types of relationship instead of 59 proposed in the task. Getting insights on the potential drop in performance induced by an increase in the size of the state space would be interesting.
>
> A6: The larger belief space could compromise the performance. Thus, we didn’t consider all 59 relationships in belief initialization and made a trade-off between efficiency and accuracy. Investigating the potential drop in performance as more relationships are used would be interesting.
>
> Q7:
> > 200 expert trajectories are used in the training set. Having an idea of the impact of the number of trajectories on the performance of the methodology could be of great interest.
>
> A7: This will certainly affect the results. However, 200 expert trajectories, compared to Li et al. [2], are already significantly smaller. And our performance is considerably better. We are happy to conduct additional experiments in the revisions to enrich our conclusions.
>
> Q8:
> > The two main limitations of the methodology are the non admissible outputs produced by the model and the back and forth behavior. The two were acknowledged by the authors.
>
> A8: We clarify that these are two cases of actions generated by the heuristic policy, which is caused by LLM policy errors that are similar to hallucinations. We admit that it affects the performance of the MCTS, but it is not LLM-MCTS that generates back-and-forth behaviors and non-admissible outputs. We will revise this part of the paper to make it clear.
>
> Our contribution is to improve decision-making. We leverage MCTS to base decisions on LLMs' world model knowledge rather than fully rely on the LLM policy. The world model of LLM could also be incorrect but is updated with observations as the agent takes action in the real world, making it more accurate over time. In addition, MCTS looks ahead multiple steps, allowing it to correct some of the errors in the set of actions proposed by the LLM during the search process. This is the reason why we outperform the GPT3.5 policy.
>
> [1] X. Puig et al, “Watch-and-help: A challenge for social perception and human-ai collaboration,” ICLR 2021.
>
> [2] S. Li et al., “Pre-trained language models for interactive decision-making,” Neurips 2022.
>
> [3] W. Huang et al. "Language models as zero-shot planners: Extracting actionable knowledge for embodied agents." ICML 2022.
>
> [4] Jang, Youngsoo, et al. "Monte-carlo planning and learning with language action value estimates." ICLR. 2020.

---

> > ### Comment · Reviewer_RKS5 · 2023-08-21
> >
> > Thanks a lot for providing a detailed answer.
> > - Adding the code in the additional material would have been helpful to better understand the way you implemented the proposed methodology. I think it is problematic not to release it during review.
> > - As acknowledged by the authors, the methodology doesn't scale to a large belief state. Additional experiments with more relationships would be important to highlight these limitations. As of now, I think this doesn't make it unsuitable for real world applications.
> > Given the two limitations, I don't consider currently increasing the score. Making the code available for review and adding experiments with larger belief states could  change the current score.

---

> > > ### Author Response · Authors · 2023-08-21
> > >
> > > Thank you for your reply.
> > >
> > > > I think it is problematic not to release it during review.
> > >
> > > As for the code, we have sent the code to the AC via a private message, which should be available after AC puts it up.
> > >
> > > > As acknowledged by the authors, the methodology doesn't scale to a large belief state. Additional experiments with more relationships would be important to highlight these limitations.
> > >
> > > * Review misunderstood our point. We acknowledge that PUCT struggles with large belief space (in Table 2, LLM-MCTS Uniform State Prior). However, the reviewer seemed to miss out on the fact that we use LLM to provide a prior that effectively narrows down the belief space for search in a large domain, which is reflected in our ablation study (Table 2). This is a key contribution of our work that is clearly claimed in the paper.
> > > * We wish to clarify that the object relationships (on, inside) are only used to **initialize** the world's belief. Using an imperfect but reasonable initial state belief is a trade-off between efficiency and accuracy. In addition, as the agent navigates and receives new observations, its beliefs of states and relationships of objects will be updated. Our experimental result suggests that the trade-off is sufficient.
> > > * Besides, this trade-off is also well applied in prior work [1], which is also used by Li et al. [2] in data collection. Even they do not use LLM to initialize the belief.
> > >
> > > > Given the two limitations, I don't consider currently increasing the score. Making the code available for review and adding experiments with larger belief states could change the current score.
> > >
> > > If the additional experiment is your **primary concern** that leads to strong rejection rather than an interesting point, clearly stating it in your initial review comment would be helpful and constructive. We cannot finish the experiment in such a short period near the discussion deadline. We will add the additional experiments in the revisions. Our initial plan to make the code available for the camera-ready version is also valid according to NeurIPS policy. If you find any part of our presentation regarding the technical details unclear, please state it.
> > >
> > > We wish to know whether our response clarifies the other issues you raised.
> > >
> > > [1] X. Puig et al, “Watch-and-help: A challenge for social perception and human-ai collaboration,” ICLR 2021.
> > >
> > > [2] S. Li et al., “Pre-trained language models for interactive decision-making,” Neurips 2022.

---

### Official Review · Reviewer_jShB · 2023-07-08

**Soundness:** 3 good
**Presentation:** 4 excellent
**Contribution:** 3 good
**Rating:** 6
**Confidence:** 3

**Summary:**

This work demonstrates that LLMs can be used as the commonsense models of the world and serve as the heuristic policy in search algorithms. Specifically this paper uses Monte Carlo Tree search to explore word states sampled from the output of LLMs and commonsense policy from LLMs can effectively guide the search, which reduces the search complexity. Experimental results on daily planning tasks again verify the advantages over using LLMs solely as policies.


**Strengths:**

a). Novel idea by incorporating commonsense knowledge from LLMs into search algorithms instead of LLMs solely as policies. This work also points out that doing the search with the help of LLMs as a model may be better and more efficient than using LLMs directly as a policy.

b). Comprehensive evaluation of the proposed methods including simple tasks, compositional tasks or in-distribution, out-of-distribution tasks.

c). Good insights that improvements might come from the MCTS’s look-ahead search and more explorations of other possible search directions. Overall this work might motivate more research utilizing LLM’s world knowledge for decision-making problems.


**Weaknesses:**

a). This work argues that planning policy may suffer from hallucination issues of LLMs in the related work. However this work did not justify how their proposed methods can help relieve the issues. It can add some context to discuss this.

b). As discussed in the paper, the proposed method might face the efficiency problem.


**Questions:**

(a) Just curious about why the baseline of UCT achieves zero success rate in Table 1 and can you explain more about this? Due to huge search space, UCT might fail to solve in a given time limit. But are there any improved search methods that can achieve better success rates, i.e., stronger baselines?


**Limitations:**

n.a.

---

> ### Author Rebuttal · Authors · 2023-08-09
>
> Thank you very much for your valuable feedback. We will carefully revise our paper to incorporate your suggestions. The following are our responses to your questions.
>
> Q1:
> > This work argues that planning policy may suffer from hallucination issues of LLMs in the related work. However this work did not justify how their proposed methods can help relieve the issues. It can add some context to discuss this.
>
> A1:  We leverage MCTS for model-based reasoning, thus basing decisions on LLMs' world model knowledge rather than being fully dependent on the policy. Assuming that the world model is correct, MCTS then deduces the decision through search. In this sense, our approach improves decision-making over incorrect predictions that are similar to hallucinations.
>
> The world model of LLM could also suffer from hallucination, but is updated with observations as the agent takes action in the real world, making it more accurate over time. In addition, MCTS looks ahead multiple steps, allowing it to correct some of the hallucinations in the set of actions proposed by the LLM during the search process.
>
> We will revise the manuscript and add this discussion.
>
> Q2:
> > As discussed in the paper, the proposed method might face the efficiency problem.
>
> A2: Our study shows the trade-off between accuracy and efficiency and explores the feasibility of our approach. See the full answer in the **runtime performance** of our global responses.
>
> Q3:
> > Just curious about why the baseline of UCT achieves zero success rate in Table 1 and can you explain more about this? Due to the huge search space, UCT might fail to solve in a given time limit. But are there any improved search methods that can achieve better success rates, i.e., stronger baselines?
>
> A3: This intractability stems from the large state and action spaces coupled with sparse rewards. These factors result in a wide and deep search tree, with the size growing exponentially and leading to intractable planning. Even though MCTS [4] is among the top online planners in POMDP, it suffers from these complexities, much like other SOTA model-based methods such as DESPOT [5], which we also tried. Due to time constraints, we are unable to include all methods in our formal experiments.
>
> [1] C.-Y. Hsieh et al. "Distilling step-by-step! outperforming larger language models with less training data and smaller model sizes." 2023.
>
> [2] C. Liang et al. "Less is more: Task-aware layer-wise distillation for language model compression." ICML 2023.
>
> [3] K. Shridhar et al. "Distilling reasoning capabilities into smaller language models." ACL 2023.
>
> [4] D. Silver et al. "Monte-Carlo planning in large POMDPs." NeurIPS, 2010. Link: https://proceedings.neurips.cc/paper_files/paper/2010/file/edfbe1afcf9246bb0d40eb4d8027d90f-Paper.pdf
>
> [5] S. Adhiraj et al. "DESPOT: Online POMDP planning with regularization." NeurIPS, 2013. Link: https://proceedings.neurips.cc/paper/2013/file/c2aee86157b4a40b78132f1e71a9e6f1-Paper.pdf

---

### Official Review · Reviewer_3x3m · 2023-07-09

**Soundness:** 3 good
**Presentation:** 3 good
**Contribution:** 3 good
**Rating:** 6
**Confidence:** 3

**Summary:**

This paper introduces a technique to incorporate large language models' commonsense knowledge into Monte Carlo Tree Search to guide planning. It uses LLMs to obtain probabilities over the initial belief of the state, and as a heuristic policy to guide simulation. Evaluation on household object rearrangement tasks in VirtualHomes demonstrates that the method is empirically more effective than search algorithms that don’t incorporate LLMs (UCT), or just having LLMs generate a policy through direct prompting (GPT3.5 Policy). The paper also makes a theoretical argument in favor of using model-based methods rather than model-free methods with LLMs.

**Strengths:**

1. Novel approach to planning with LLMs that doesn’t just entail having the LLM directly generate a policy, but as part of a search algorithm
2. Technique significantly outperforms both vanilla search algorithms and using LLMs directly to generate policies on household object rearrangement tasks
3. Ablation study and analysis give good insight into what helps, and where there is still room for improvement


**Weaknesses:**

1. The paper evaluated this technique in only occurred in one domain -- VirtualHome -- and it is unclear whether it can generalize to other embodied domains.
2. The main barrier to using this approach in practice is that it entails making multiple calls to GPT3.5 when computing each action, alongside having to do additional search on top -- making it less efficient than both vanilla search and vanilla LLM generation methods. Though computational expense was noted as a limitation in the conclusion, it may also be valuable to additionally report runtimes in the paper
3. The theoretical arguments for “knowledge of LLMs regarding states of world is more complete than their knowledge of policies for accomplishing daily tasks” (L58-59, Section 5.3), whereby description length of policies vs. world models is used to justify this claim, is not entirely convincing to me. First, it is unclear whether the argument the authors put forth that “learning about the world would likely require less training data than learning policies to do all tasks in the domain” (L62) applies to LLMs, which have been trained on an abundance of data and are not at all limited by data scarcity. This argument also makes assumptions about the way LLMs represent policies vs. models which seem not at all obvious to me — why is description length the best way to characterize LLM knowledge? LLMs are not trained to be efficient compressors, they are trained to imitate the training set.
    * Is there a way to empirically validate this claim in particular?

---
Missing citations:
-  https://aclanthology.org/2022.acl-long.120/

**Questions:**

1. Instead of performing similarity search, can you just constrain generation and/or prompt the model explicitly to choose amongst a limited set of available actions/object? This may help avoid translation errors.
2. Can you clarify the connection between LLM knowledge and description lengths?

**Limitations:**

Some limitations were mentioned in conclusion, though there is no explicit limitations section.

---

> ### Author Rebuttal · Authors · 2023-08-09
>
>
> Thank you very much for your valuable feedback. We are grateful for the many suggestions for improvement, which we will incorporate in the revised manuscript. We would like to further clarify the questions and concerns you raised.
>
> Q1:
> > The paper evaluated this technique in only occurred in one domain -- VirtualHome -- and it is unclear whether it can generalize to other embodied domains.
>
> A1: We chose VirtualHome, as it is an established domain well used in prior work [1-3]. Our method is domain-agnostic and should be able to generalize to other common household domains, as LLMs have vast general knowledge that should be widely applicable [4-6]. We will explore those distinct datasets for further evaluation.
>
> Q2:
> > The main barrier to using this approach in practice is that it entails making multiple calls to GPT3.5 when computing each action, alongside having to do additional search on top -- making it less efficient than both vanilla search and vanilla LLM generation methods.
>
> A2: There is a trade-off between accuracy and computational efficiency. While our method requires multiple LLM calls, it provides substantially improved results (Table 1). One objective of our study is to identify this possibility and highlight the trade-off. There are also various ways to enhance runtime performance, such as using a smaller LLM like Llama or distilling domain knowledge into a smaller model [7-9]. We are keen to explore these avenues in future research.
>
> Q3:
> > it may also be valuable to additionally report runtimes in the paper
>
> A3: The actual runtime to finish MCTS and make one decision depends on the number of simulations and internet connection latency. We used 100 times simulation during the tree search for GPT3.5-MCTS in our experiments, and it takes 1 to 2 minutes on average to make one decision. We will report the details in the paper and appendix in the final version.
>
> Q4:
> > Can you just constrain generation and/or prompt the model explicitly to choose amongst a limited set of available actions/object? This may help avoid translation errors.
>
> A4: We have tried various methods to restrict the action generation. We have put the list of pre-defined actions and observed object list in the prompt. However, there are still remaining errors in LLM’s policy, such as opening the fridge when it is still far away. This is likely because the LLM cannot guarantee the effective use of all the information and preconditions in the prompt when making decisions.
>
> Q5:
> > Can you clarify the connection between LLM knowledge and description lengths?
>
> A5: The assumption that LLM's training data is unlimited and can cover all domains and tasks is something we respectfully disagree with. While LLM is trained on vast datasets encompassing many disciplines, it's not guaranteed to cover all situations in all possible tasks. An extreme but representative example is large-number multiplication. If we wish to learn the policy, we need to have a dataset that covers all possible multiplications of two numbers and remember the results. It is impossible to have such a dataset as there are an infinite number of cases. On the other hand, describing a world model (0-9 digits and multiplication rule) is much simpler. Therefore, LLM does have data scarcity issues for learning the policy of some tasks.
>
> The description length, reflecting the space complexity to represent full knowledge, determines the data needed to cover situations in a task or domain. It is often used in learning theory to analyze sample complexity, e.g., see Chapter 2 of Kearns and Vazirani (link: https://www.cis.upenn.edu/~mkearns/teaching/CIS625/KearnsVaziraniChapter2.pdf). Our analysis in the paper shows that the world model may require far less training data to learn than the policy in some situations, and our experiments show that this may happen in common household tasks. LLM essentially imitates the dataset. Thus, when the data is limited, the LLM’s knowledge of the world model is likely more complete than the policy.
>
> [1] I. Singh et al., “ProgPrompt: Generating Situated Robot Task Plans using Large Language Models”, ICRA 2023.
>
> [2] S. Li et al., “Pre-trained language models for interactive decision-making,” Neurips 2022.
>
> [3] Huang et al. Language Models as Zero-Shot Planners: Extracting Actionable Knowledge for Embodied Agents, ICML 2022.
>
> [4] T. Silver et al. Generalized Planning in PDDL Domains with Pretrained Large Language Models. 2023.
>
> [5] B. Liu et al. "Llm+ p: Empowering large language models with optimal planning proficiency." 2023.
>
> [6] S. Bubeck et al. "Sparks of artificial general intelligence: Early experiments with gpt-4." 2023.
>
> [7] C.-Y. Hsieh et al. "Distilling step-by-step! outperforming larger language models with less training data and smaller model sizes." 2023.
>
> [8] C. Liang et al. "Less is more: Task-aware layer-wise distillation for language model compression." ICML 2023.
>
> [9] K. Shridhar et al. "Distilling reasoning capabilities into smaller language models." ACL 2023.

---

> > ### Comment · Reviewer_3x3m · 2023-08-14
> >
> > Thank you for your response. I believe the authors have adequately addressed my concerns regarding generalizability of the method and efficiency, and I believe the paper will be stronger with results on other household datasets and runtimes being reported.
> >
> > I also appreciate the authors detailed response on their description length argument, which has been very clarifying. While I buy the argument for the example(s) presented in the paper and find the assertion that "world models are easier to learn than policies" makes intuitive sense, I believe it would still be good to ground the theoretical argument a little more in experiments, especially as it seems to be more of an illustrative example (which makes certain assumptions) than a formal proof. At the very least, it would be good to affirm that GPT3.5 truly has stronger priors over household room-object-container relations vs. household task policies. Overall, I believe this would be a very interesting argument to make with important implications for this LM + decision-making area of research, which is why it would be good to consolidate it even more with further evidence.
> >
> > However, given that this is not the central contribution of the paper, I do not believe it necessarily affects my overall recommendation, which still leans positive.

---

> > > ### Author Response · Authors · 2023-08-14
> > >
> > > Thank you for your reply! Your suggestions will greatly strengthen our work and improve our manuscript's presentation. We have added the missing citation in our related work as well.

---

> > > ### Author Response · Authors · 2023-08-21
> > >
> > > Thank you very much for your suggestion to ground the statement "LLM has more comprehensive knowledge about world modeling than policy" more firmly in experimental results. We conducted further experiments accordingly.
> > >
> > > We conducted experiments about planning for air travel from a starting city to a destination city, which we analyzed in our introduction. We utilized GPT-3.5 to generate flight paths between cities. We compare it to the GPT-3.5 model-based approach: we use GPT-3.5 to predict neighboring cities connected by a direct flight, which feeds into the uniform-cost search (i.e., replace node expansion by GPT-3.5 as the world model).
> > >
> > > We use the data from the Kaggle World cities database, select 62 cities with populations exceeding 5 million in different countries, and use the Virtual Radar Server to get the flight routes dataset as ground truth. In our tests, we sampled 200 city pairs, evaluating path accuracy by verifying each direct flight exists. Paths were accepted if all flights were valid, even if they extended beyond our selected 62 source and target cities.
> > >
> > > The preliminary result suggests that 50.4% of the paths predicted by the GPT-3.5 policy are correct, while the GPT-3.5 world model + shortest path algorithm achieved 63.5%. Our findings reasonably supplement that LLMs’ knowledge exhibits more comprehensiveness for world modeling than policy. It will serve as a side finding to supplement our central argument.
> > >
> > > Due to time constraints, we can only finish these experiments within a very short period. We will add this result and further experiments to the appendix in the final manuscript if accepted.

---

### Author Rebuttal · Authors · 2023-08-09

# Global response

We thank all the reviewers' efforts invested in reviewing our work and providing valuable feedback. We summarize the main concerns raised by reviewers and our corresponding responses.

One review (RKS5) states that
> The paper mostly relies on experiments and ideas from S.Li 2022. and would gain clarity if clearly stated. For instance, using LLMs to initialize the belief state was already used in S.Li 2022.

We believe this is a severe misunderstanding and mischaracterization of our method. Our research utilizes GPT-3.5 as a commonsense world model for **model-based search** with MCTS, while Li et al. [1] use behavior cloning to finetune GPT-2 as a **model-free policy**. There are fundamental differences between the model-based and the model-free approaches, conceptually and algorithmically. In particular, the power of the model-based approach lies in its ability to compose elementary pieces of (commonsense) world knowledge through a reasoning/planning procedure (MCTS, in our case) and achieve compositional generalization. This is often one key weakness of the model-free approach. We made this argument in the introduction as well as in Sect 5.3. We will try to further clarify this issue during the revision.

Further,  unlike what the review  (RKS5) asserts, Li et al.'s approach [1], by its nature, does not track belief states.

In responding to the main weaknesses raised by the reviewers, we appreciate the opportunity to address the concerns:
* **The domain and task of the experiments**: Reviewers noted that our experiments are restricted to object rearrangement in the VirtualHome simulator. Object rearrangement is a representative embodied AI task [3-9] with many practical implications in everyday life, such as setting the table, tidying up the room, loading the dishwasher, and more. Thus, object rearrangement experiments are an interesting setting to investigate a fairly large set of planning capabilities required in embodied AI. We chose VirtualHome, as it is an established domain well used in prior work [1,2,18,19]. Our method is domain-agnostic and should be able to generalize to other common household domains, as LLMs have vast general knowledge that should be widely applicable [15-17]. We will explore those distinct datasets for further evaluation.
* **Runtime performance**: There is a trade-off between accuracy and computational efficiency. While our method requires multiple LLM calls, it provides substantially improved results (Table 1). One objective of our study is to identify this possibility and highlight the trade-off. There are also various ways to enhance runtime performance, such as using a smaller LLM like Llama [10,11] or distilling domain knowledge into a smaller model [12-14]. We are keen to explore these avenues in future research.

We thank the reviewers for their constructive criticism, and we hope our rebuttal has clarified the issues.

[1] S. Li et al., “Pre-trained language models for interactive decision-making,” NeurIPS 2022.

[2] X. Puig et al., “Watch-and-help: A challenge for social perception and human-ai collaboration,” ICLR 2021.

[3] B. Dhruv et al., “Rearrangement: A Challenge for Embodied AI.” 2020.

[4] L. Weihs et al. "Visual room rearrangement." CVPR 2021.

[5] A. Szot et al. "Habitat 2.0: Training home assistants to rearrange their habitat." NeurIPS 2021.

[6] Y. Kant et al. "Housekeep: Tidying virtual households using commonsense reasoning." ECCV 2022.

[7] A. Khandelwal et al. "Simple but effective: Clip embeddings for embodied ai." CVPR 2022.

[8] E. Huang et al.,  "Large-scale multi-object rearrangement." ICRA 2019.

[9] A. Krontiris et al. "Dealing with Difficult Instances of Object Rearrangement." RSS 2015.

[10] H. Touvron et al. "Llama: Open and efficient foundation language models." 2023.

[11] H. Touvron et al. "Llama 2: Open foundation and fine-tuned chat models." 2023.

[12] C.-Y. Hsieh et al. "Distilling step-by-step! outperforming larger language models with less training data and smaller model sizes." 2023.

[13] C. Liang et al. "Less is more: Task-aware layer-wise distillation for language model compression." ICML 2023.

[14] K. Shridhar et al. "Distilling reasoning capabilities into smaller language models." ACL 2023.

[15] S. Bubeck et al. "Sparks of artificial general intelligence: Early experiments with gpt-4." 2023.

[16] T. Silver et al. Generalized Planning in PDDL Domains with Pretrained Large Language Models. 2023.

[17] B. Liu et al. "Llm+ p: Empowering large language models with optimal planning proficiency." 2023.

[18] I. Singh et al., “ProgPrompt: Generating Situated Robot Task Plans using Large Language Models”, ICRA 2023.

[19] Huang et al. Language Models as Zero-Shot Planners: Extracting Actionable Knowledge for Embodied Agents, ICML 2022.

---

### Author Response · Authors · 2023-08-21
**Reply to all reviewers**

As the discussion period ends soon, we briefly summarize the main review comments along with our responses.

The reviewers have highlighted several key strengths in our research:
* **Novel Approach**: All reviewers acknowledge and appreciate the novelty in our planning approach with LLM, particularly in its utilization as part of a search algorithm rather than using LLM solely as a policy.
* **Better Performance**: Reviewers 3x3m, jShB, Mf2f, and euNw highlighted our method's superior performance over existing techniques in object rearrangement tasks.
* **Comprehensive Evaluation**: Reviewers jShB and RKS5 appreciate our thorough evaluation, encompassing a broad range of tasks, including simple tasks, compositional tasks, and both in-distribution and out-of-distribution tasks.
* **Insightful Ablation Study**: Reviewers 3x3m, jShB, RKS5, and Mf2f appreciate our ablation study and analysis for providing good insights into the effective components of our methodology and areas where there is room for improvement.

We have made detailed responses to clarify the reviewers’ concerns:
* One review (RKS5) misunderstood our paper’s main contribution and stated that our paper relies mainly on ideas from Li et al. [1]. Our research utilizes GPT-3.5 as a commonsense world model for model-based search with MCTS, while Li et al. [1] use behavior cloning to finetune GPT-2 as a model-free policy. There are fundamental differences between the model-based and the model-free approaches, conceptually and algorithmically. Details are presented in our global response.
* **Domain and tasks in experiments**. We mainly solve object rearrangement tasks as it has many useful and practical purposes in daily life, and our experiments investigate a fairly large set of planning capabilities required in embodied AI. Virtualhome is also a well-established platform used in many prior works. See the full answer in our global response.
* **Runtime performance**. Our study shows the trade-off between accuracy and efficiency and explores the feasibility of our approach. We also introduced various potential methods that could enhance the runtime performance, which is interesting for future research. See the full answer in our global responses.
* **LLM’s world model v.s. LLM policy**. Reviewer 3x3m suggests we ground "LLM has more comprehensive knowledge about world modeling than policy" more firmly in experimental results. We conducted additional experiments about planning for air travel from a starting city to a destination city, which we analyzed in our introduction. Our preliminary result suggests that GPT-3.5’s world model + shortest path algorithm achieved 63.5% accuracy, higher than the 50.4% accuracy achieved by the GPT-3.5 policy. It is a side finding of our paper that supplements the central argument. See the full answer in our reply to the reviewer 3x3m.
* **large belief state**. In the latest reply message, reviewer RKS5 misunderstood our point and stated that our method cannot scale up to a large state belief. Classical methods, such as PUCT (in Table 2, LLM-MCTS Uniform State Prior), struggle with large belief states. However, RKS5 seemed to miss out on the fact that we use LLM to provide a prior that effectively narrows down the belief space for search in a large domain, which is reflected in our ablation study (Table 2). This is a key contribution of our work that is clearly claimed in the paper. Details are in our reply message to RKS5.
* **Code**. Reviewer RKS5 wishes to see our code. We have shared our implementation with the AC via a private message.

[1] S. Li et al., “Pre-trained language models for interactive decision-making,” Neurips 2022.

---

### Decision · Program_Chairs · 2023-09-21

**Decision:**

Accept (poster)

**Comment:**

The paper introduces how LLMs can be used to accelerate search. It shows that they can both be initial models of the world and heuristics to guide search with MCTS. The results are algorithm is demonstrated with household rearrangement tasks in VirtualHome.

The reviewers generally found the paper clear and novel, and the results impressive and thoroughly ablated. Generally, the reviewers were positive with the exception of RKS5, whom after a detailed discussion in rebuttal is weak reject. As such I recommend acceptance.

Still, there are a few improvements that should be made for a final version, primarily cleaning up the code, finishing the air travel experiments, and a few clarifications. The clarification are in regards to (1) the first figure and (2) generally the use of world model throughout.

For (1), showing where the LLM is used would be informative and generally more algorithmic information, as the current one is so sparse it has essentially no information value. Showing each place the LLM is used very clearly would be great, e.g., showing the LLM output the initial state distribution given the task.

For (2), it should be more clearly stated many times (including the abstract, the intro) that the world model in this case is just the initial belief state rather than the transition and new states, which are instead handled by the environment. It would also help to have a pseudocode algorithm box, under ten lines (maybe five) in the method section that tracks with a figure to very clearly deliver the message.